# Shallow subsurface heat recycling is a sustainable global space heating alternative

Susanne A. Benz [1✉], Kathrin Menberg[2], Peter Bayer [3] & Barret L. Kurylyk [1✉]

Despite the global interest in green energy alternatives, little attention has focused on the large-scale viability of recycling the ground heat accumulated due to urbanization, industrialization and climate change. Here we show this theoretical heat potential at a multi-continental scale by first leveraging datasets of groundwater temperature and lithology to assess the distribution of subsurface thermal pollution. We then evaluate subsurface heat recycling for three scenarios: a *status quo* scenario representing present-day accumulated heat, a *recycled* scenario with ground temperatures returned to background values, and a *climate change* scenario representing projected warming impacts. Our analyses reveal that over 50% of sites show recyclable underground heat pollution in the *status quo*, 25% of locations would be feasible for long-term heat recycling for the *recycled* scenario, and at least 83% for the *climate change* scenario. Results highlight that subsurface heat recycling warrants consideration in the move to a low-carbon economy in a warmer world.

[1] Dalhousie University, Centre for Water Resources Studies, Halifax, Nova Scotia, Canada. [2] Karlsruhe Institute of Technology, Institute of Applied Geosciences, Karlsruhe, Germany. [3] Martin Luther University Halle-Wittenberg, Department of Applied Geology, Halle, Germany. ✉email: susanne.benz@dal.ca; barret.kurylyk@dal.ca

In 1950, 30% of the world's population was settled in an urban environment, but by 2018, this number rose to 55% and is projected to increase further[1]. This ongoing global urbanization concentrates energy demands in highly populated areas[2], and further results in local to regional waste heat accumulation. Although research on urban heat islands and anthropogenic heat flows has predominantly focused on or above the land surface[3,4], the shallow subsurface also continuously absorbs and stores heat[5–10]. Consequently, subsurface temperatures, which are typically measured in shallow groundwater, are elevated under urban heat islands and in most places affected by human activity[11–13]. Due to their sheer number and ubiquity, heat lost from buildings and heat from sealed surfaces warmed by solar radiation are the main contributors to large-scale subsurface warming in a stable climate[14–17]. While less ubiquitous, even a single component of buried infrastructure can elevate ambient ground temperatures by several degrees Celsius[18]. Despite widespread subsurface thermal pollution, combining underground infrastructure with shallow geothermal energy systems to recycle their waste heat is far from widespread, and while recycling the accumulated heat of urban (or even rural) settlements has been studied for selected locations[19,20], its global potential is unclear.

The relative lack of attention paid to large-scale subsurface heat recycling (i.e., extraction of the additional shallow subsurface heat from urbanization, industrialization, and climate change) as a green energy solution with potential for global climate change mitigation is presumably due in part to a lack of awareness of decision makers, budgeting and technical concerns, and the absence of large-scale studies. However, this low-carbon approach to space heating warrants more attention as (1) it is reliable and independent of weather and time of day (unlike solar or wind power), (2) heat production and demand are physically collocated removing the need for costly transportation between the two, (3) it does not disturb the connectivity and functioning of aquatic systems (unlike tidal energy or hydropower from dammed rivers[21]), and (4) not recycling the accumulating heat will result in further underground thermal pollution with potential consequences such as declining groundwater quality[22–24] and adverse effects on groundwater-dependent ecosystems[25,26]. Also, while the contribution of subsurface urban heat islands to atmospheric temperatures has not yet been quantified, it is likely that cooling the underground will play at least a marginal role in mitigating urban warming[27]. More importantly, shallow subsurface heat recycling is a sustainable and renewable heating source and can therefore reduce this sector's carbon footprint[28,29]. This is particularly relevant as heating is the dominant residential energy use in many locations: 60% and 75% of the residential energy

consumption are used for space and water heating in the US[30] and Europe[31], respectively. Among the standard technologies used for extracting heat from the shallow ground are open-loop systems such as groundwater wells that circulate extracted groundwater through heat pumps. If groundwater cannot be easily extracted due to relatively dry or impermeable ground, closed-loop systems such as borehole-heat exchangers, energy collectors, or piles are applied to circulate a heat carrier fluid instead. Given the rising need for air conditioning in summer[32], we want to highlight that these technologies are also able to provide cooling. Moreover, where feasible aquifer thermal energy storage makes it possible to recycle summer's heat in winter[33], further increasing the potential for subsurface heat recycling discussed herein.

The objective of this study is to quantify the theoretical feasibility of subsurface heat recycling at a multi-continental scale, focusing on its energetic sustainability and renewability under present and future climate conditions without consideration of any technical constraints. Based on groundwater temperatures measured at more than 8000 locations in Europe, North America, and Australia, we investigate this feasibility for three different scenarios (Fig. 1): (1) The *status quo* represents present-day conditions in which heat has accumulated in the shallow subsurface, particularly in areas shaped by human activity. (2) A scenario for which the accumulated heat has been *recycled*. In this scenario, groundwater temperatures have returned to undisturbed levels, which facilitates a higher heat input from the surface into the underground. (3) Lastly, we assess the impact of *climate change* on our *recycled* scenario: groundwater temperatures remain at their undisturbed level, while surface temperatures increase in accordance with climate change projections. This results in rising heat input from the surface and a concurrent reduction in heating demands. Lastly, we identify locations in Europe currently most feasible for recycling heat accumulated in the shallow subsurface.

## Results

**Status quo.** In many locations worldwide, shallow subsurface temperatures are elevated due to accumulated heat from infrastructure, climate change, and land cover/land use changes. We quantified the volumetric density of excess thermal energy (Fig. 2) as the product of heat anomalies (the difference between local groundwater temperatures and median rural background groundwater temperatures) and ground volumetric heat capacity (see Methods for a detailed description of parameters and equations).

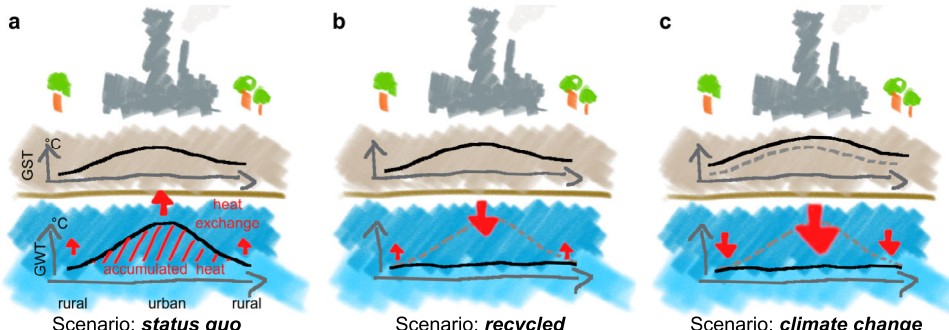

**Fig. 1 Schematic drawing of the analyzed scenarios. a** The *status quo* describes elevated groundwater temperatures (GWTs) under the built environment that result in accumulated heat (see Eq. (2)). **b** The *recycled* scenario highlights heat exchange (see Eq. (4)) between the surface and aquifer after the accumulated heat has been recycled and GWTs are restored to their undisturbed levels. **c** The impacts of climate change increase ground surface temperatures (GSTs) and thus impact the heat demand and the heat exchange between the surface and aquifer. We apply this scenario on top of the *recycled* scenario.

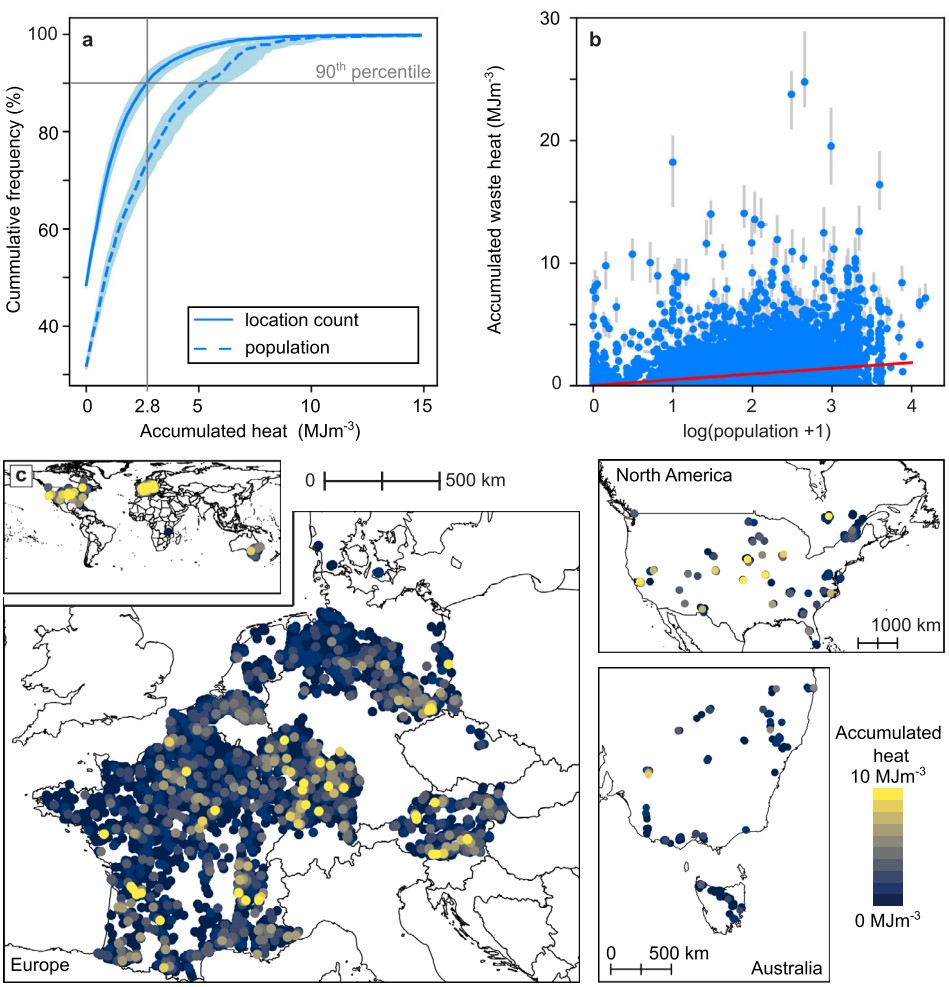

**Fig. 2 Accumulated heat for the *status quo*. a** Cumulative distribution of the heat accumulated per cubic meter at our 8118 study sites. The blue line indicates the median values, while our maximum and minimum estimates are colored in light blue. **b** The accumulated heat increases with population density. The red linear fit projects an increase of 0.46 MJ m$^{-3}$ per tenfold increase in population. Error bars depict our maximum and minimum estimates. **c** Maps displaying the accumulated heat. A zoomable version of this map is accessible under https://susanneabenz.users.earthengine.app/view/feasible-heat-recycling.

Only 44% of our sites ($n > 8000$, mostly rural) are not characterized by heat pollution, and we find accumulated heat up to 2.8 MJ m$^{-3}$ (90th percentile, 2.5 MJ m$^{-3}$ based on our lowest estimates of input parameters and 3.2 MJ m$^{-3}$ based on the highest estimates; Fig. 2a). For the remainder of this study all results will be given in the form *results main analysis (results lowest estimates, results highest estimates)*. After weighting for population density, we find that 71% (68%, 75%) of people live where the study sites that have accumulated shallow subsurface heat, and 26% (27% following highest and lowest estimates of heat capacity and population densities) live where accumulated heat is above the 90th percentile for the total dataset (Fig. 2a). This is further demonstrated in Fig. 2b, which reveals that accumulated heat scales with population. Variations in local ground conditions (i.e., volumetric heat capacity) have little effect on spatial differences in the amount of accumulated heat which is instead predominantly driven by the magnitude of the groundwater temperature anomalies (Supplementary Fig. 1). Accordingly, the map of accumulated heat is a blueprint of urbanization and vice versa (Fig. 2c), implying that the distribution of heat is generally focused where heating demand is also highest.

We argue that this accumulated heat could and should be recycled across much larger scales. Such recycling is typically performed at smaller spatial scales with technologies (e.g., groundwater heat pumps or borehole heat exchangers) that perform optimally in permeable ground that facilitates groundwater flow. Hence, as a conservative approach that is focused on productive porous aquifers rather than fractured rock environments, we filter out all locations without unconsolidated sediments (i.e., keeping only locations with the soil types *sand and coarser*, *sand and silt or clay*, *silt and/or clay*, or *mixed* following Supplementary Table 1; leaving us with just above 6000 out of the initial > 8000 locations). We note that even in these unconsolidated sediments local conditions may vary, and not all heat is recoverable based on today's technology and land availability. In this study, we use the subset of locations with unconsolidated sediment whenever we compare our results to heating demands, but work with the entire data set when quantifying accumulating heat. To quantify the feasibility of subsurface heat recycling, we compare the local annual space heating demands, estimated from heating degree days, population density, and GDP[34], to the accumulated heat (expressed as a volumetric density) in Fig. 3a. Diagonal lines indicate the necessary vertical extraction interval to meet 1 year's heating demand through the energy generated when recycling the accumulated heat with a geothermal system and heat pump with

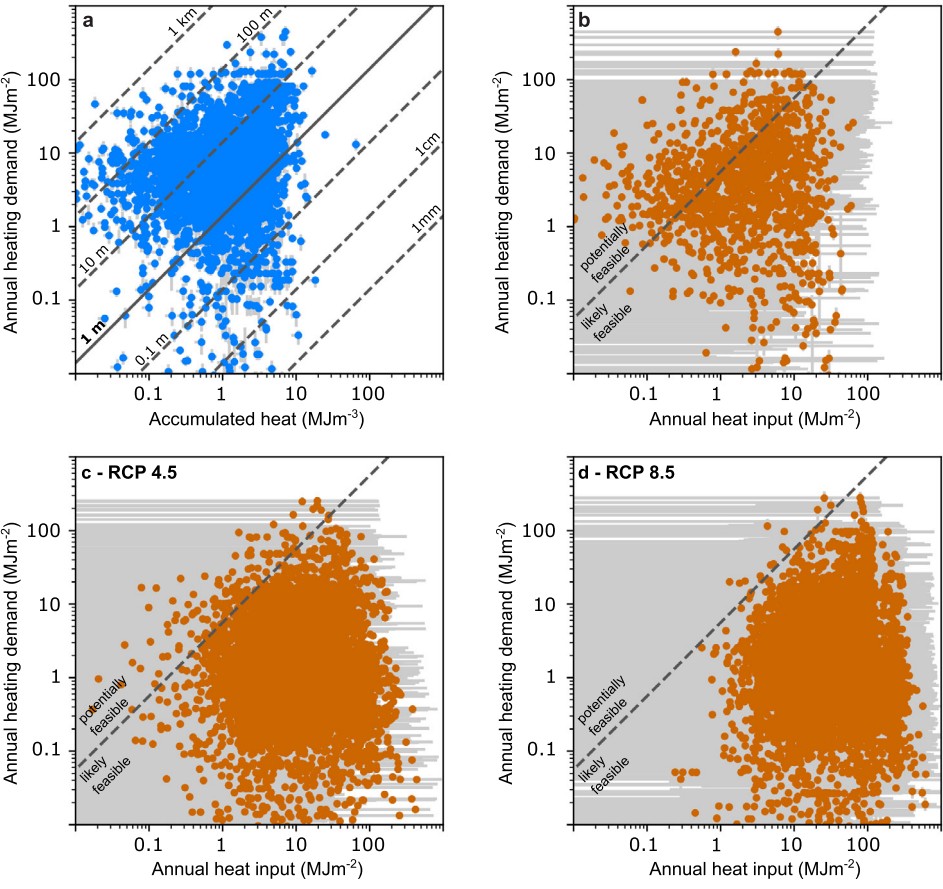

**Fig. 3 Feasibility of subsurface heat recycling under all scenarios.** Please note: for all parts of this figure only locations with unconsolidated sediments are shown. **a** Comparison of accumulated heat of the *status quo* and annual heating demand. The diagonal contour lines indicate the extraction length necessary to generate one year of heat when recycling the accumulated heat. (Note that locations with no heating demand or no accumulated heat are not shown). Comparison of annual mean heat input and heating demand for the *recycled* scenario (**b**), and the scenario *climate change* with RCP 4.5 (coupled with the shared socioeconomic pathway (SSP) 2 (**c**), and RCP 8.5 (coupled with SSP 5, **d**). The diagonal line separates *likely feasible* locations (i.e., the generated heat is more than 25% of the annual heating demand) from *potentially feasible* locations (i.e., the generated heat is less than 25% of the annual heating demand). (Note that locations with no heating demand or no heat input are not shown).

a COP of 3.5. For example, 12% (9%, 15%) of the sites are located below the 1 m line, which means that recycling the accumulated heat of an only 1 m thick zone of permeable ground could supply at least an entire year's worth of heat (Fig. 3a). However, accumulated heat is typically distributed over much more than 1 m of aquifer thickness, and is observed up to 100 m depth (decreasing with depth; e.g.,[35]). Hence, extracting heat over a longer interval is typically recommended. For 50% (47%, 52%) of all locations an extraction interval of 20 m or less will suffice to supply 1 year or more of heat (for 43% (39%, 47%) 10 m are sufficient; Supplementary Fig. 2a). However, as the extraction interval should be located in the aquifer, the required borehole length needs to be longer (extraction interval plus depth to groundwater table, Supplementary Fig. 2b). Still, for 43% (0%, 49%) of all locations a borehole length of 20 m can generate enough heat for 1 year; for 29% (0%, 41%) 10 m will suffice. The high uncertainties are a product of the high uncertainties in the depth to the groundwater table, where we assume values of ±10 m for our low and high estimates.

This subsurface heat could provide a carbon-reduced source of thermal energy, but it is not endless. For this energy source to be sustainable and renewable, the extraction rate must not exceed the rate of surface heat that is lost to the underground. Assuming energy extraction is focused on the groundwater table depth, this

balance is achieved when the extraction rate is equivalent to the sum of conductive heat fluxes between the groundwater table and the natural and built environment at the surface. These fluxes have previously been determined for selected cities[14,19,36,37]. In three of these studies local infrastructure such as district heating networks and subsurface parking garages dominate on the smaller scale. However, over larger areas the studies indicate that buildings and elevated temperatures caused by sealed surfaces are dominant contributors to subsurface heat accumulation. Therefore, we focus here on heat exchange between the groundwater table and buildings and/or the land surface, with positive values indicating downward heat flow.

The annual heat exchange calculated from groundwater, surface, and building temperatures and considering groundwater table depth and building density, show that for 79% (97%, 16%) of the more than 8000, mainly rural locations, the subsurface presently loses heat to the surface; however, values overall range from −56.5 to +2.7 MJ m$^{−2}$ (10th–90th percentile; −158.8 to −5.5 MJ m$^{−2}$ for the lowest estimates of input parameters and −0.8 to 32.7 MJ m$^{−2}$ for the highest estimates; Fig. 4a). Again, the high uncertainties here and in the following stem from uncertainties associated with the depth to the groundwater table. While these low or negative numbers are expected for undisturbed (rural) heat exchange where the negative values

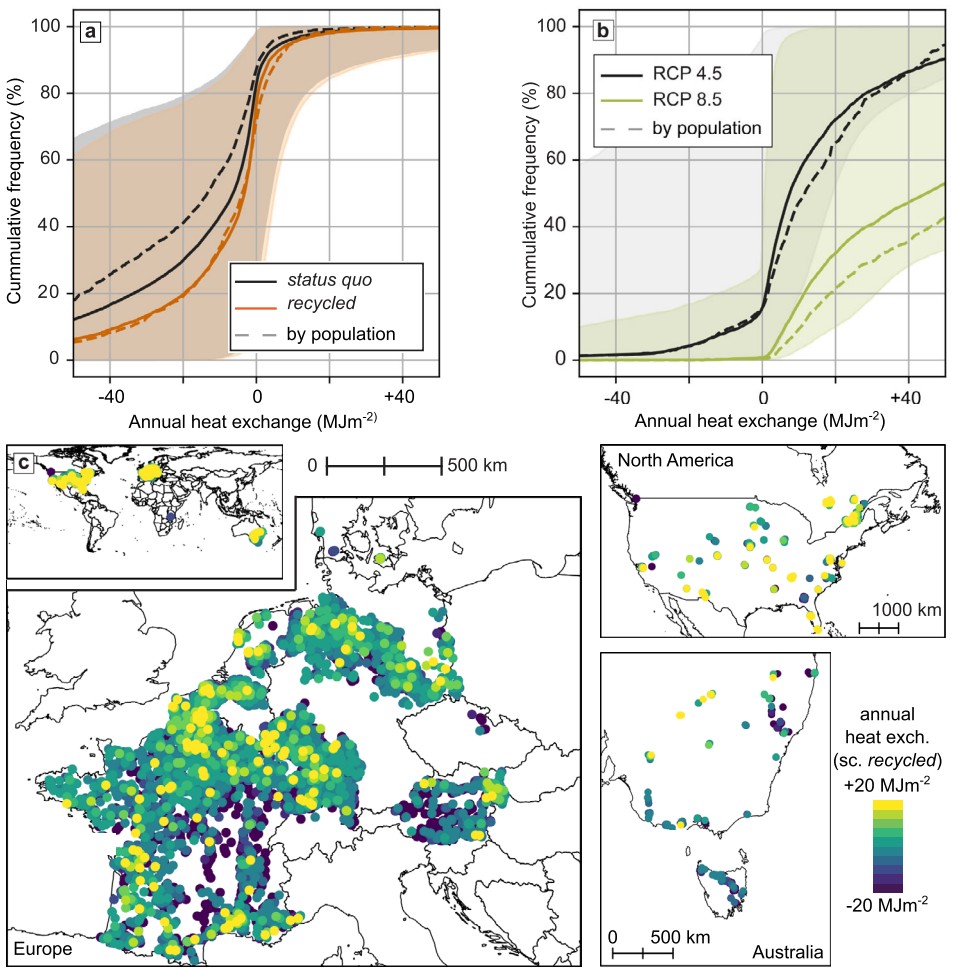

**Fig. 4 Annual mean heat exchange under all scenarios. a** Cumulative distributions of heat exchange between the groundwater table and the surface and/ or buildings for the *status quo* (black) and *recycled* (orange) scenarios. Positive values indicate heat input into the aquifer. The shaded areas indicate our maximum and minimum estimations. Dashed lines indicate population-weighted distributions. To keep the figure clear, maximum and minimum population-weighted estimations are not shown. **b** Cumulative distributions of heat exchange between the groundwater table and the surface and/or buildings in 2095–2100 following RCP 4.5 (black) and RCP 8.5 (green). Shaded areas indicate our maximum and minimum estimations. Dashed lines indicate population-weighted distributions. To keep the figure clear, maximum and minimum population-weighted estimations are not shown. **c** Maps of the annual mean heat exchange for the *recycled* scenarios. A zoomable version of this map, as well as the map for the scenarios under *climate change* are accessible under https://susanneabenz.users.earthengine.app/view/feasible-heat-recycling.

arise from the geothermal heat flux from the interior of the earth, we find that heat flow from the subsurface to surface is greatest (i.e., heat exchange is most negative) in areas where most subsurface heat has already accumulated (Supplementary Fig. 3a). This indicates that in the *status quo* scenario, the subsurface heat reservoirs are overflowing for these locations and the shallow subsurface is no longer a heat sink to the surface. Although accumulated heat and population correlate (Fig. 2b), we find no reverse correlation between heat exchange and population (Supplementary Fig. 3b), implying that it is disproportionately the more suburban or even rural thermally polluted groundwater reservoirs that have reached their limits and generate heat flow to the (comparatively) cold surface. While these are less common than heated urban aquifers, they have been identified as more thermally polluted in previous studies[12].

Following our analysis of accumulated heat, we quantify the feasibility of recycling this heat by comparing heating demands and the annual heat input (Supplementary Fig. 4) for the subset of data with unconsolidated sediments. We sort locations into *not feasible* (no heat input, i.e., heat loss, or no heating demand —82%, of all locations in unconsolidated soils), *potentially*

*feasible* (generated heat from recycling the heat input is less than or equal to one quarter of the annual heating demand—4%), or *likely feasible* (generated heat from recycling the heat input is larger than one quarter of the annual heating demand—14%). Based on our lowest estimates of input parameters 99% of locations are *not feasible* due to the assumed distance to the groundwater table. Based on our highest estimates 14% or *not feasible*, 7% are *potentially feasible*, and 78% are *likely feasible*. Supplementary Fig. 4 further compares our results to existing studies quantifying both heating demand and heat input[14,15,19,20,37]. All of these are focused on larger cities and hence have high heating demands. In addition, all but the study focusing on Osaka (which resulted in the lowest annual heat input)[37] explicitly quantify small-scale heat sources such as underground infrastructures commonly found in city centers. Comparison indicates that locally heat input from small-scale sources might be up to one order of magnitude higher than the one from the large-scale sources we quantify. This indicates that for the *status quo* scenario it is critical to check for local heat sources when assessing the feasibility of a small-scale subsurface heat recycling system.

**Status once accumulated heat has been recycled.** The computed magnitude of surface-aquifer heat exchange is largely driven by the distance between the land surface and groundwater (i.e., water table depth), and the associated temperature difference (Supplementary Fig. 1). Thus, lowering groundwater temperatures increases the local heat exchange such that more heat is deposited in the aquifer. This is clearly visible in the *recycled* scenario in which the accumulated heat has been recycled and groundwater temperatures have been returned to their undisturbed levels (Fig. 1b) while land surface and building temperatures remain disturbed. In this scenario, aquifers in 28% (3%, 91%) of the locations (compared to 21% (3%, 84%) in the *status quo*) have heat input from buildings and/or elevated surface temperatures (Fig. 4a—the figure shows the reverse: 72 or 79% have negative heat exchange) and heat exchange is overall higher, ranging from -37.0 to +4.1 MJ m$^{-2}$ (10th–90th percentile; −125.8 to −4.5 MJ m$^{-2}$ for the lowest estimates of input parameters and +0.2 to 35.4 MJ m$^{-2}$ for the highest estimates; Fig. 4a). There is no significant correlation between heat exchange in this scenario and accumulated heat for the *status quo* scenario (Supplementary Fig. 3a) or population density (Supplementary Fig. 3b). We see two reasons for this lack of correlation: (a) the rising temperatures of the past decades lead to heat input in places outside the built environment, and (b) buildings and elevated surface temperatures are not the only drivers of accumulated heat in the *status quo*, instead it is enhanced by local sources such as factories, industry parks, landfills or similar[12].

If we do not continuously reduce or recycle the annual heat input, locations with a positive heat exchange in this *recycled* scenario would begin accumulating heat again. Hence, extracting the annual heat input each year would not only be the most sustainable and renewable use of this heat, it is also the only approach to protect water quality and groundwater dependent ecosystems from subsurface thermal pollution. By comparing local heat input (i.e., heat input once the accumulated heat has been recycled) to heating demands (Fig. 3b) for our subset of data in unconsolidated sediments, we find that for 18% of all locations (housing 10% of the population), subsurface heat recycling is *likely feasible*, i.e., the heat generated when extracting the annual heat input is more than one quarter of annual space heating demands. For another 7% (housing 20% of the population), subsurface heat recycling is *potentially feasible* (i.e., the heat generated is between 0 and 25% of the heating demand), whereas the remaining locations have either no heat input or no heating demand. In our minimum estimates, 99% of locations are *not feasible* and 1% *potentially feasible*; in our maximum estimates 6% are *potentially feasible* and 87% *likely feasibly*—at least based on current surface temperatures.

**Status under climate change.** Under climate change scenarios, surface temperatures will increase and shift the heat exchange between surface and aquifer in a positive direction (i.e., more heat input into the aquifer). While this heat input would not transport anthropogenic waste heat as it is not caused directly by structures such as buildings and streets, it can still be recycled from the subsurface as an ongoing low-carbon energy source and as a means to maintain present groundwater temperatures. In our final scenario *climate change*, we thus combined the effects of the *recycled* scenario with surface warming due to climate change following the RCP 4.5 or 8.5 pathways of CMIP5 (Fig. 1c). Due to a lack of data for future building projections, we neglect changes in building density. These analyses project that by the end of the century (2095-2099) following RCP 4.5, 85% (4%, 100%) of all our locations (99% (72%, 100%) in RCP 8.5) will have heat input to the subsurface from increased surface temperatures and/or buildings, and overall ranges

are elevated to −5.2 to +49.0 MJ m$^{-2}$ (−155.0 to −3.7 MJ m$^{-2}$, +7.0 to +64.0 MJ m$^{-2}$) for RCP 4.5 and +6.7 to +125.7 MJ m$^{-2}$ (−49.2 to +4.3 MJ m$^{-2}$, +20.5 to +476.9 MJ m$^{-2}$) for RCP 8.5 (10th to 90th percentile, Fig. 4b).

Heat input is then compared to space heating demands for the same time period (2095–2099) for our subset of data in unconsolidated sediments. In addition to the impact of increased temperatures, these demands are also adjusted for projected population densities following the shared socioeconomic pathways (SSPs)[38,39]. We project that the feasibility of subsurface heat recycling will increase drastically (Fig. 3c and d) due to the combined effects of increased rate of heat flow from the surface and decreased heating demands in a warmer world. Following RCP 4.5, only 16% (98%, 0%) of our locations are *not feasible* for subsurface heat recycling, 2% (1%, 0%) are *potentially feasible*, and 81% (1%, 100%) are *likely feasible* (i.e., the heat generated when sustainably extracting the heat input is more than one quarter of space heating demands). In fact, for 73% (0%, 98%) of our locations the generated recycled heat is equal to or larger than local heating demands. For RCP 8.5, 99% (53%, 100%) of all locations are *likely feasible* with 97% (30%, 100%) projected to generate at least as much heat as the annual space heating demands when extracting the annual heat input.

**Mapping the feasibility of subsurface heat recycling.** So far, we have only assessed selected locations with available groundwater temperature data. We will now expand on the results of our sites to map the feasibility of subsurface heat recycling in Europe where more than 90% of our sites are located. For this analysis we expand on the *recycled* scenario which identifies suitable locations for present-day sustainable and renewable subsurface heat recycling (Fig. 5a). This is achieved by training a random forest classifier on 70% of all suitable locations based on heating demand, building density, depth to the groundwater table, thermal conductivity, and ground surface temperatures. Results are shown in Fig. 5b and accessible as a zoomable map under https://susanneabenz.users.earthengine.app/view/feasible-heat-recycling. Following the concerns discussed above, only areas of unconsolidated sediments are considered.

The overall accuracy based on the confusion matrix of validation data is 0.81 (Supplementary Fig. 5). In total 350,000 km$^2$ were classified. Of those, 90% (housing 74% of the population) are classified as *not feasible*, 2% (housing 23% of the population) as *potentially feasible*, and 7% (housing 2% of the population) as *likely feasible*. While overall population, groundwater temperatures, groundwater table depth, and space heating intensity (i.e., the amount of thermal energy necessary to heat 1 m$^2$ per heating degree day) have the most impact on feasibility based on our locations (Supplementary Fig. 1), the map highlights that urban centers are often classified as only *potentially feasible*, whereas less densely populated regions are *likely feasible* due to lower heating demands. We must note that our analysis gives values, including heating demand, as a spatial density—hence in rural areas with a low population density, one household would extract the heat input over a large area to fulfill their heating needs.

## Discussion

We provide the first global evidence that recycling subsurface heat accumulating due to climate change and urbanization is a feasible and sustainable green alternative to conventional space heating methods for many locations, and its feasibility will increase drastically over the next century. Like the thermal pollution in subsurface urban heat islands, warming climates at the surface will contribute to underground heat reservoirs where the

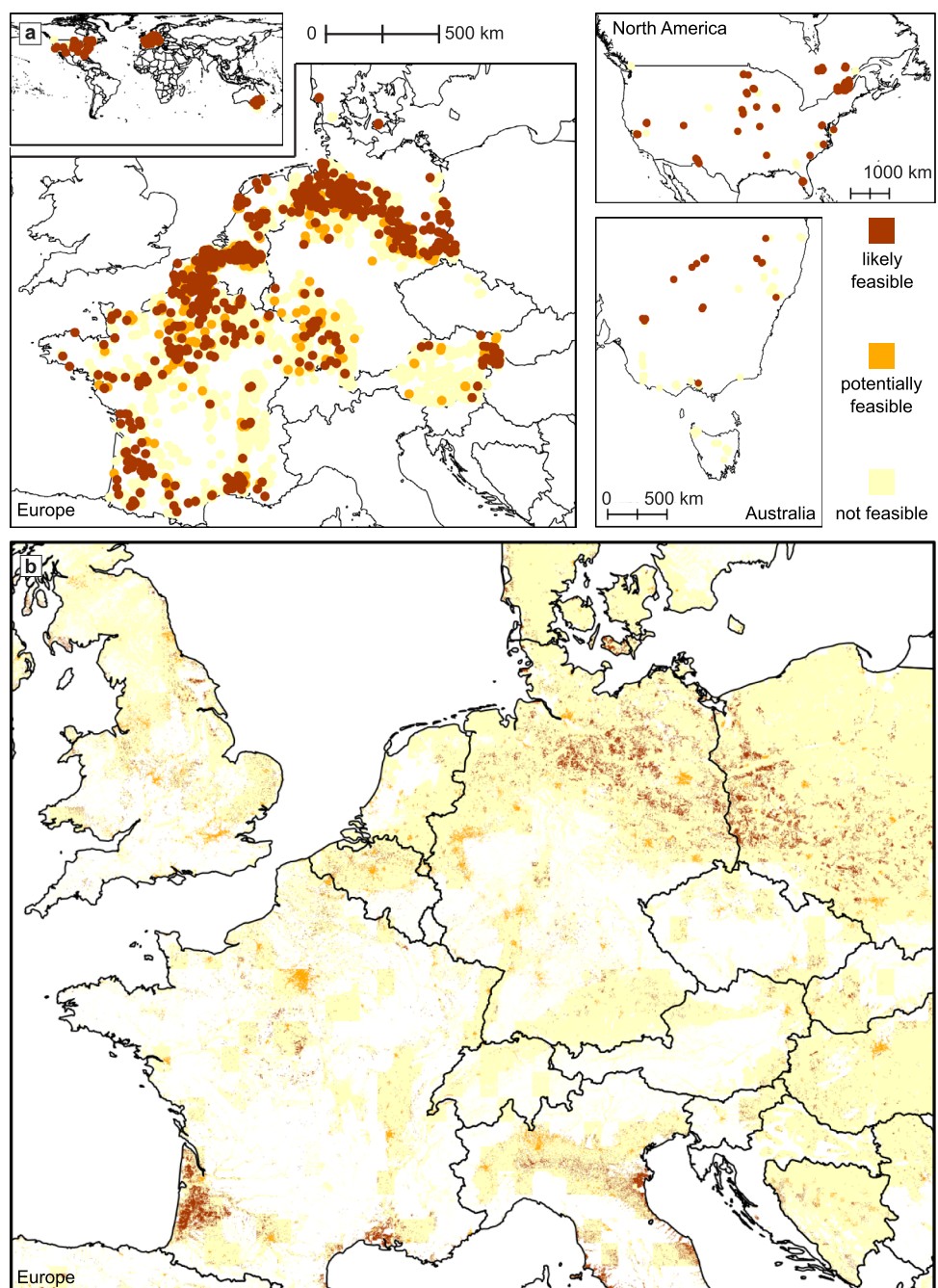

**Fig. 5 Feasibility of subsurface heat recycling for our *recycled* scenario.** Please note: for all parts of this figure only locations with unconsolidated sediments are shown, or rather classified. **a** Map displaying the feasibility of recycling the annual heat input. **b** Results of the random forest classification. A zoomable version of these maps is available from https://susanneabenz.users.earthengine.app/view/feasible-heat-recycling.

excess heat can be recycled with shallow geothermal energy systems. The ability to utilize this theoretical geothermal potential will depend on the accessibility and extractability of the ground heat for geothermal systems. Subsurface heat recycling will allow us to capitalize on warming climates, while actively mitigating future climate change through a low-carbon heating approach and reducing the adverse effects of rising temperatures on our (ground-)waters.

Over the past few decades heat has accumulated in the underground where it might impact groundwater quality and groundwater-dependent ecosystems, including surface water bodies with thermal regimes and thermal heterogeneity influenced by groundwater inputs. More than 55% of the 8000+ locations studied here are thermally polluted—this does not consider the effects of our warming climate but rather describes local waste heat, predominantly from anthropogenic sources. Assuming a shallow (20 m) borehole depth, extracting this excess heat could generate one year of low-carbon heat for 43% of all locations. This heating source becomes almost carbon-free when the heat pump is also supplied with electricity from other sustainable sources.

If we cannot block (e.g., through insulation) the annual subsurface heat input from the surface and buildings, heat will continue to accumulate, unless it is continuously recycled. Based on our estimate, 7% of the European study area can supply more than 25% of the annual heating demand by recycling the annual

heat input. An additional 2% of the area (housing 23% of the European population) could generate up to 25% of the annual heating demand. It is important to note that this is in addition to the extractable geothermal heat flux that is of similar magnitude (on average $2\,MJ\,m^{-2}$ [40]) and the recyclable heat input from potential local heat sources such as underground infrastructures which can outperform the quantified heat input from buildings and the surface by more than an order of magnitude[15,19,20]. Also, our analysis does not consider the increased viability of ground heat recycling for areas where ground heat pumps are also used for space cooling in warm months. Therefore, the figures presented for heat recycling are highly conservative, and they reveal an enhanced shallow geothermal energy potential that is overlooked when relying on natural conditions alone.

We also showcase the option to recycle the heat input from climate change - in contrast to other common methods of renewable energy supply[41], the feasibility of subsurface heat recycling will increase over time. Depending on the scenario RCP 4.5 or RCP 8.5, 73% or 97% of all studied locations would be able to fulfill their annual heating demand by the end of the century by simply recycling the heat input from the surface and buildings given the additional downward heat flux due to climate warming. Should policymakers and stakeholders decide against shallow geothermal heat recycling this heat will continue to accumulate in the underground adversely affecting water quality and ecosystems.

## Methods

**Groundwater temperatures and their anomalies**. This study is based on 8118 data points derived from mean (multi-)annual groundwater temperature (GWT) measurements in 10,168 locations listed in Supplementary Tables 2 and 3. Data were primarily collected from government agencies for the time period 2000–2015 and filtered for quality following the examples of previous studies[12,42]. This includes focusing on temperature data measured at depths less than 60 m. To confirm that (multi-)annual means are not seasonally biased, we also filtered out wells with a seasonal radius $r$ larger than 0.25, where $r$ is a unitless measure describing if a well has been observed uniformly over the seasons ($r = 0$) or if all measurements have been taken in the same month ($r = 1$)[42]. While groundwater temperature measurements were collected all over the world, this rigorous filtering resulted in (multi-)annual mean groundwater temperatures being primarily observed in Europe (86% of all locations). All other data are from North America (10% of all locations), Australia (3% of all locations), or Tanzania (5 locations). It is furthermore important to note that these wells are clustered and are not spread out evenly through the regions. This is most obvious for locations in the US, where we for example, were not able to collect a single data point in Oregon, but in other places find several GWT measurement locations within 1 km². Temperatures range from 0.3 °C in northern Quebec, Canada to 37.9 °C near a geothermal hot spot in Austria. Due to the mostly shallow depths, we expect little impact from the geothermal gradient and assume depth-independent annual mean temperatures. A map is presented in Supplementary Fig. 6 and a cumulative histogram in Supplementary Fig. 7.

Groundwater temperature anomalies $\Delta T$ are determined by subtracting rural background temperatures from local temperatures (compare[11,12]). $\Delta T$ is also commonly used as a measure of urban temperature anomalies at the surface (e.g., [43,44]) and indicates how much warmer, or colder, a location is compared to the nearby rural background conditions. While designed to quantify urban heat, this method can be applied to all locations, regardless of land use, to quantify local temperature anomalies of anthropogenic and natural causes. It is determined as

$$\Delta T_i = T_i - median(T_{i,rural}) \tag{1}$$

where $T_i$ is the (multi-)annual mean groundwater temperature of location $i$, and $T_{i,rural}$ are the temperatures of all rural locations in the background of location $i$. Here, following the established guidelines of previous studies[11,12], the rural background is defined as all locations (including potentially location $i$) within 47 km of location $i$, of a similar elevation (±90 m) as location $i$, and with a nighttime light of DN14 (Digital Number) or less. To run this analysis on the collected data, elevation was extracted from the Global Multi-resolution Terrain Elevation Data 2010[45]. Data on nighttime lights were compiled from Version 4 of the DMSP-OLS Nighttime Lights Time Series, Image and Data processing by NOAA's National Geophysical Data Center, and DMSP data collection by the US Air Force Weather Agency. Following the time-period that GWTs were collected, we determined the 15-year mean from 2000 to 2015. Both values, elevation and nighttime light, were extracted for each location through Google Earth Engine (GEE)[46] at a 1000 m scale.

To ensure a minimum of statistical meaning, only wells with at least three rural background locations are included, leaving us with 8932 locations. $\Delta T$ ranges from −4.9 °C at a spring near the Grand Canyon, USA to 27 °C at the well near the geothermal hotspot in Austria. The highest temperature anomaly $\Delta T$ outside natural occurring hot spots is 9.5 °C in a well in an industrial complex near Grenoble, France. See Supplementary Figs. 6 and 7 for maps and a histogram. In these histograms and in our main analysis, only 8118 locations (90% of those in Europe) are discussed for which all data used in this study could be extracted.

Note that this analysis neglects the vertical characteristics of anthropogenic temperature anomalies. They are commonly highest near the surface and decline with depth, as the generating heat source (e.g., a building or underground infrastructure) is typically found at or near the surface.

**Accumulated heat**. Accumulated heat describes the thermal energy stored in the shallow underground caused by climate change, urbanization and other anthropogenic heat sources (e.g., underground infrastructure such as tunnels). Here, we determine it for the *status quo* for local aquifers per cubic meter (heat density, $Q$) by multiplying the mean volumetric heat capacity $c_V$ of the aquifer with the estimated heat anomaly $\Delta T$ :

$$Q = \begin{cases} \Delta T \cdot c_V & |_{\Delta T > 0} \\ 0 & |_{\Delta T \leq 0} \end{cases} \tag{2}$$

To determine the volumetric heat capacity of the aquifer we predominantly rely on the global unconsolidated sediments map database[47]. More precisely, we extract grain size information, and—where that is not available—sediment class which we relate to soil type following previous studies[48]. If both of those are not available, we rely on the global lithological map database[49]. This methodology for classification has been previously suggested for global permeability maps[48]. Here, we assign to each class heat capacities based on the VDI 4640 guidelines[50] (Supplementary Table 1, Supplementary Fig. 7). These guidelines give a range of values for each soil type. We use the minimum and maximum of that range for our minimum and maximum analysis, and the mean of these two values for our main analysis. Where our soil types consist of two rock types described in the VDI 4640, we take the mean of these rock types. Furthermore, for unconsolidated sediments we use the values given for the water-saturated soil type to best represent aquifer conditions.

The method quantifying accumulated heat was developed from previous studies (e.g., [51]) and—like those—determines Q for minimum, maximum and median heat capacities $c_V$. It has no way of separating accumulated anthropogenic waste heat from natural thermal anomalies such as hot springs. As discussed, when introducing $\Delta T$, we find the highest values near a hot spring in Austria. We decided not to filter our data set for geothermal activity, as a) there is no clear indicator of where to do so, and b) while thermal hot springs are not accumulated waste heat per se, they still represent an increased potential for geothermal heat extraction and with this provide a sustainable local heating alternative.

**Building density**. Building footprints in the United States are available as vector data released by Microsoft in 2018. Depending on the underlying imagery, capture dates average around 2012 but vary greatly. These footprints were later rasterized and released as building density[52].

To extrapolate building density to our global $\Delta T$ data set we relate them to median nighttime light and built-up land cover fraction for all 7,970,663 1 km × 1 km pixels in the United States. The built-up land cover fraction is extracted from the Dynamic Land Cover map of 2015 (the first year available) released by the Copernicus Global Land Service[53]. A linear fit reveals that building density $\rho_{Bld}$ is a function of built-up land cover fraction $BUF$, and nighttime lights $NTL$.

$$\rho_{Bld} = 0.11 + 0.003 \cdot BUF \cdot NTL \tag{3}$$

The coefficient of determination $r^2$ is 0.73 (Supplementary Fig. 8). Several alternative regressions were tested without improvements (Supplementary Table 4). Built-up land cover fraction was exported for all locations in a 1000-m scale through GEE. A histogram of estimated building density can be found in Supplementary Fig. 7.

**Annual heat exchange**. The annual heat input into the underground has previously been determined for selected urban locations[14,19,36,37], and results indicated that the dominant large-scale heat sources are elevated ground surface temperatures and buildings. Although at smaller scales, local infrastructure such as district heating networks and subsurface parking garages have significantly more impact on local temperatures, these data are not available on the global scale. Hence, we focus on the annual heat exchange $q$ between the aquifer and the surface ($q_{surf}$) and/or buildings ($q_{bld}$). We use the directionally ambiguous term exchange instead of input, to highlight that many of our locations are not in an urban environment, and hence aquifers lose heat to the surface and not the other way around. Annual heat exchange for the *status quo* at our locations is determined as:

$$q = (1 - \rho_{bld}) \cdot q_{surf} + \rho_{bld} \cdot q_{bld} \tag{4}$$

where the building density $\rho_{bld}$ is expressed as the percent of area covered by buildings.

Following Fourier's law we quantify the conductive heat transport from the surface to the groundwater table depth ($D_{GW}$) as:

$$q_{surf} = \lambda \frac{T_{surf} - GWT}{D_{GW}} \qquad (5)$$

and from buildings as:

$$q_{bld} = \lambda_{bld} \frac{T_{bld} - GWT}{D_{GW}} \qquad (6)$$

with the thermal conductivity $\lambda$ of the unsaturated zone and, if applicable, buildings ($\lambda_{bld}$), ground surface temperature $T_{surf}$ and building temperature $T_{bld}$.

While some of the collected GWTs measurement location also report groundwater table depth, most do not. Hence, we need to rely on available global assessments[54,55]. These data are the results of a model run at hourly time steps for the decade 2004 to 2014 and are made available in a 30 s grid. For the updated model[55] no evaluation was published, hence we follow the one from the original data to set minimum and maximum estimates[54]. The standard deviation of residuals is at about 10 m; we therefore set maximum and minimum table depth at ±10 m. In addition, these data underestimate the observed water table depth in temperate climates by about 1.5 m, potentially due to widespread pumping lowering observation data[54]. Without corrections, 45% of our locations are assigned a groundwater table depth of less than 50 cm and 73% less than 150 cm. To account for this underestimation, we conservatively set a minimum of 1.5 m for minimum and/or median groundwater table depth. For compatibility with all other parameters used in this study, we implemented these data into GEE and extracted values for all location in a 1000 m scale (Supplementary Fig. 7).

The process to determine thermal conductivity is analogous to the one determining heat capacity (Supplementary Table 1, Supplementary Fig. 7). However, here the VDI 4640 guidelines give recommended values which we use in our main analysis. For our minimum and maximum analysis we still rely on the ranges given in the guidelines. As heat transport is through the unsaturated zone, for unconsolidated sediment we use values recommended for a moist environment, or where these are not given, the mean of the recommendations for dry and water-saturated conditions. Additionally, for $\lambda_{bld}$ the unsaturated zone is adjusted in the first 45 cm to represent the thermal properties of the underfloor of the building and thermal conductivities are combined using the harmonic mean. For these first 45 cm of the unsaturated zone, we set median (minimum, maximum) $\lambda$ to $\lambda_{underfloor}$ of 0.16 (0.02, 1.6) W m$^{-1}$ K$^{-1}$ modeling concrete with 10 cm glass wool insulation, an air filled underfloor, or a concrete floor without insulation (compare[37]).

As an estimate of ground surface temperatures, we exported annual mean soil temperature at 0–7 cm depth from the ERA5-Land monthly average reanalysis product[56] (native resolution of 9 km) for each year between 2000 and (including) 2014. It is important to note that effects of urban heat islands are not depicted well in this reanalysis, and temperatures in the urban environment might be underestimated. Annual means are exported for all locations in a 1000-m scale through Google Earth Engine. These annual mean data are later summarized to maximum, minimum, and median (Supplementary Fig. 7).

Building temperature was estimated as 23.5 (18, 27) °C[57]. These values represent operative temperature standards for the design and assessment of energy performance of buildings as suggested by the International standards ISO 17772. Maximum and minimum values are based on summer and winter for the indoor environmental Class III: Moderate (will still provide an acceptable environment). Our median values are based on Class I: High and combine summer and winter as well as residential and non-residential areas.

Besides annual heat exchange of the *status quo* we also determine the annual heat exchange once the accumulated heat has been recycled (*recycled* scenario). This includes using the above Eqs. (5) and (6) with GWTs minus their anomalies:

$$GWT_{recycled} = \begin{cases} GWT - \Delta T & |_{\Delta T > 0\,°C} \\ GWT & |_{\Delta T < 0\,°C} \end{cases} \qquad (7)$$

Heat exchange is determined for maximum, minimum, and median values.

**Heat exchange at the end of the century**. To assess the impact of climate change on this heat exchange, we repeat our analysis of annual heat exchange for the *recycled* scenario (Eqs. (4)–(7)) but base $T_{surf}$ on projected shallow ground surface temperatures at the end of the century following the scenarios RCP 4.5 and RCP 8.5 of the CMIP5 program.

To determine projected ground surface temperatures, we followed the guidelines for 1 cm soil temperatures set by previous work[58] and herein focused on mean temperatures between Jan 2095 and Dec 2099. We also only worked with the models BCC-CSM1-1, BNU-ESM, CanESM2, CCSM4, INM-CM4, IPSL-CM5A-LR, MIROC5, MPI-ESM-LR, MRI-CGCM3, and NorESM1-M as they were suggested by the authors[58] and are available as part of the downscaled air temperature scenarios for the globe[59] used later in this study to determine future heating demands. Data were collected from the World Climate Research Program at https://esgf-node.llnl.gov/search/cmip5/. For comparability with other parameters, mean temperatures of each model were implemented in GEE and extracted in a 1000-m scale for all locations.

We must note that our future scenarios for the heat exchange do not consider possible changes in building density and insulation. To our knowledge, no

meaningful projections of these parameters are currently available. Hence our data give the projected potential heat exchange based on projected surface temperatures ($T_{surf}$ is $T_{surf,RCP4.5}$ or $T_{surf,RCP8.5}$ in Eq. (5)), but without changes in infrastructure (no change in $\rho_{bld}$ and $\lambda_{bld}$ in Eqs. (4) and (6)).

**Heat generated from subsurface heat recycling**. Like all heat pumps, ground source heat pumps convert mechanical work into heat and supplement this with the pumped (i.e., recycled) heat. The generated thermal energy is therefore the sum the recycled heat and additional heat from electric power consumption. Hence, we define the heat generated from heat recycling as

$$Q_{generated} = Q \cdot \nu \qquad (8)$$

with the accumulated heat (for the *status quo*) or the annual heat input (for the *recycled* and *climate change* scenarios) as Q. The efficiency $\nu$ is related to the coefficient of performance COP with $\nu$ = COP/(COP-1). While COP varies for different models, it typically ranges between 3 and 4[28] and has been set to 3.5 in this study ($\nu$ = 1.4).

Due to technical constraints shallow geothermal systems are primarily installed in areas with porous soil and dynamic groundwater flow. Hence, only locations in unconsolidated sediments are chosen (Supplementary Table 1) when discussing the heat generated from subsurface heat recycling. Aside from these, fractured aquifers could be considered and these would broaden the area of application substantially, but due to the challenges in mapping fracture networks and determining permeabilities they are not covered in the conservative approach here.

**Heating demand**. Heating demand $Q_{demand}$ is determined following previous works[34] as

$$Q_{demand} = HDD \cdot A \cdot I \cdot pop \qquad (9)$$

with heating degree days $HDD$, the average living space per person $A$, the space heating intensity $I$, and population density $pop$. This heating demand is then used for the *status quo* and *recycled* scenarios.

UN-adjusted population density for the years 2000, 2005, 2010, and 2015 were extracted from the Population of World Version 4.11 Model (native resolution of 1 km)[60] for all locations through GEE in a 1000-m scale (Supplementary Fig. 7). Data are summarized as maximum, minimum and median population densities.

The space heating intensity is an empirical value indicating the amount of thermal energy necessary to heat 1 m$^2$ per heating degree day. Values are estimated by country[61] and listed in Supplementary Table 5. To estimate the average living space per person we follow an existing regression linking them to the GDP[34]:

$$A = 6.22 \cdot \log_2(GDP) - 28.95 \qquad (10)$$

GDP estimates of 2014[62] are again listed in Supplementary Table 5 by country.

Heating degree days (HDDs) with a base temperature $T_B$ of 18 °C were determined for each year between 2000 and (including) 2014. HDDs are a common tool to assess heating needs following the simplified assumption that temperatures below 18 °C indicate a need for heating.

$$HDD = \sum \begin{cases} T_{i,dailymean} - T_B & |_{T_i > 18\,°C} \\ 0 & |_{T_i < 18\,°C} \end{cases} \qquad (11)$$

Daily mean air temperatures $T_{i,dailymean}$ are extracted from the same ERA5 product as ground surface temperature[56]. As ERA5 is a monthly mean data set we calculate HDD based on monthly mean air temperature and multiply the results by 30.4375 to account for the average number of days per month. We must also note again that the ERA5 reanalysis is not developed to monitor urban temperature, hence urban heat islands are not necessarily picked up.

Annual HDDs for all years are determined in GEE and exported for all locations with $\Delta T$ in a scale of 1000-m (Supplementary Fig. 7). For our analysis they are then summarized into minimum, median and maximum values.

**Heating demand at the end of the century**. For the *climate change* scenario, we project heating demands at the end of the century based on air temperature projections from the CMIP5 scenarios RCP 4.5 and RCP 8.5. Like with the projected heat exchange, we do not consider changing infrastructures (i.e., no change in average living space per person and space heating intensity). However, in contrast to the projected heat exchange we do consider changes in population density. These have a direct impact on heating demand, but only indirectly (through building density) influence heat exchange.

Heating degree days are thus calculated based on downscaled CMIP5 climate scenarios[59] RCP 4.5 and RCP 8.5 for the models BCC-CSM1-1, BNU-ESM, CanESM2, CCSM4, INM-CM4, IPSL-CM5A-LR, MIROC5, MPI-ESM-LR, MRI-CGCM3, and NorESM1-M that were also used for our projected ground surface temperatures. Similar to that analysis, we first determine mean HDDs between 2095 and (including) 2099 for each model before exporting them at a 1000-m scale for all locations from GEE and summarizing them in median, minimum, and maximum values.

Global population projection grids based on the Shared Socioeconomic Pathways (SSPs)[38,39] are provided by the socioeconomic data and applications center by decade. Here we extract data in GEE in a 1000-m scale for SSP2 *Middle of*

the Road and SSP5 *Fossil-fueled Development*. SSP5 links to RCP 8.5 in its base scenario, it is the only pathway to reach these levels of radiative forcing. We link SSP2 to RCP 4.5 but must note that it is neither the only pathway able to reach this scenario, nor can it do so without some mitigation efforts.

**Feasibility of subsurface heat recycling in Europe.** To better understand where recycling the annual heat input of the *recycled* scenario is feasible, we train a random forest classifier[63] and apply it to Central Europe. Because other regions have so few groundwater temperature data they are not assessed in this part of the study. We furthermore only focus on areas with unconsolidated sediments (following the classification of Supplementary Tables 3 and 6).

Regions are classified into one of three categories:

- *Not feasible* indicates areas where there is either no heat input or no heating demand.
- *Potentially feasible* indicates areas where the heat generated when recycling the annual input is 25% or less of the annual space heating demands.
- *Likely feasible* indicates areas where the heat generated when recycling the annual input is more than 25% of the annual space heating demands.

Summarizing variables as much as possible we use heating demand (determining it per pixel following Eq. (9)), building density (following Eq. (3)), depth to the groundwater table, thermal conductivity, and ground surface temperature to train the classifier. Histograms of all variables are displayed in Supplementary Fig. 9.

Overall, 5499 locations of our data set are suitable as training data for this analysis (i.e., are in an unconsolidated sediment in Europe). Of those 79% are classified as *not feasible*, 7% as *potentially feasible*, and 14% as *likely feasible*. We train the random forest classifier on 70% of these data, using the remaining 30% for validation. Considering the importance of training data sample selection[64], we assign probabilities based on the depth to the groundwater table at our locations—the variable with a distribution least represented by the data set. Probabilities are thus set to the ratio of the histograms for groundwater table depth of pixels and groundwater table depth of point locations. Supplementary Fig. 9 displays the cumulative histograms for all variables for all locations as well as for the selected training locations. We run the random forest classification with 200 trees. Tests ranging from 50 to 500 trees revealed only little influence with the overall accuracy (affecting only the third decimal place), in line with previous studies[65]. Overall accuracy based on the training data is 0.96, while overall accuracy based on the validation data is 0.81. With the uneven distribution of all classes (most locations have no heat supply), inaccuracy stems primarily from locations with a heat supply of zero (no heat input) being classified as low or even high supply regions (Supplementary Fig. 5).

## Data availability

A table of the 8118 locations analyzed in this study together with a shape file and .tif file of the generated map of heat supply for Europe is made available at the Scholars Portal Dataverse under https://doi.org/10.5683/SP3/2UTTVQ[66]. Please see the Supplementary Notes for a more detailed description.

## Code availability

All codes used are also enclosed at the Scholars Portal Dataverse under https://doi.org/10.5683/SP3/2UTTVQ[66]. (Jupyter Notebook (Python) and Google Earth Engine (Javascript)).

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

## Acknowledgements

S.A.B. was supported through a Banting postdoctoral fellowship, administered by the Government of Canada. B.L.K. was supported through the Canada Research Chairs program. K.M. was supported by the Margarete von Wrangell-program of the Ministry of Science, Research and the Arts Baden-Württemberg (MWK). The support for P.B. from the German Research Foundation (DFG) under grant number 2850/3-1 is gratefully acknowledged. Figures 2, 4, and 5 and Supplementary Fig. 6 show outlines of countries based on the *World Country Polygons—Very High Definition* made available in the World Bank Data Catalog. We thank Carolin Tissen for sharing data she collected in her study on groundwater temperature anomalies in Europe[12] and the many other people and agencies collecting groundwater temperature data and making it available through (publicly accessible) databases. Our thanks also go to Philipp Blum (KIT, Germany) for his helpful feedback in the development of this study.

## Author contributions

S.A.B., K.M., P.B., and B.K. designed the study. S.A.B. lead the writing, prepared all data and code for analysis, and designed all figures; K.M. prepared data and code to determine heating demands. All authors interpreted results, wrote, and edited the manuscript together.

## Competing interests

The authors declare no competing interests.
