## [Peer Review File · Nature Communications]

Shallow subsurface heat recycling: a sustainable global space heating alternativeREVIEWERS' COMMENTS

Reviewer #1 (Remarks to the Author):

Dear authors,

the manuscript presented by Benz et al. entitled "Shallow underground waste heat: a sustainable global heating resource" for Nature Communications represents a very timely and global assessment that promotes the use of currently wasted heat in conjunction with underground urban heat islands. To my opinion the manuscript only requires minor revisions, here my suggestions:

- Abstract: specify "other disturbances"; Provide more values for the potential in terms of heat demand today and in the future.
- p. 2, l. 25: specify: "potential consequences"
- p. 3, l. 49: already define "natural levels" here
- p. 10, l. 212-214: include in abstract?
- Generally: Discuss and differentiate between theoretical and technical (and further?) potentials.
- Figure 2/4/5: Enlarge world map and include zooms; enlarge a,b,c...
- Figure 5: It seems that there is a strap in northern Netherlands / Germany / Poland, is this an artifact?

All the best,
Jannis Epting

Reviewer #2 (Remarks to the Author):

This paper presents a global approach for estimating shallow subsurface heat resources, now and in the future, and assesses where they have utility to cover current and future heat demand. While global assessment come with a great set of uncertainties and caveats (one of which being that they cannot be validated), they have value in highlighting the scale of available resource potential. The methodology draws on a number of existing approaches developed by the authors, which have been combined and extended in this paper for application on the global scale. While global assessments have been done for related resources, e.g. for deep geothermal heating/ cooling, shallow heat resources have not been quantified on the global scale. As such the approach is novel and will be of interest to the readers of this journal.

The paper is very well written and the methodology is explained in sufficient detail to make it reproducible. I have some comments for consideration by the authors, as detailed below:

- (1) The paper is very well written, but having methodology at the end means that, when reading main body of this paper, reader is left wondering in some places how parameters were derived/ what was/wasn't considered etc. This can get in the way of understanding what figures show and what the finding actually mean in practical terms. I suggest to add some very brief explanations to the text, and have included some suggestions below where I think this would improve readers' understanding.
- (2) There is some confusion as to when the analysis was done on the full data sets (>8000) and when it used the subset (unconsolidated sediments) which should be clarified. See comment in detailed list below. If different data sets were used for the different analyses, it would be useful if this was highlighted in the figure captions too.
- (3) Given that this is a global assessment, the uncertainties associated with assessment itself as well as linked to assumptions/ inherent in the data sets could have been discussed in a bit more detail and how they impact the resource estimates. Uncertainties have been mentioned for some of the data

sets/parameters, but not for others. I have included examples in list below of where I think this could be added, e.g. for thermal property data.

Specific comments:

Line 79 “unconsolidated sediments” could be understood to include clays etc. I suggest to be more specific here, i.e. ...rock environments, we only include locations where unconsolidated sediments of suitable grain size are present (leaving”

Line 101 need to make clear here that this sentence refers to references in previous sentence , perhaps add “...garages were found by these studies to”

Line 105 this is where I think a bit more detail would be useful , e.g. adding something like “The annual heat exchange calculated from groundwater, surface and buildings temperatures and considering groundwater table depth and building density, show that for 79%”

Line 105 “more than 8,000 ,,locations” – was this analysis done on the full data set rather than the unconsolidated sediment subset?

Line 114 -116 “we find no reverse correlation herehave reached their limit” – Could this be due to lower air temperatures in rural areas? I.e. even though waste heat accumulation is much lower in rural areas so are air temperatures, i.e. T gradients and hence fluxes could be quite similar to urban areas? Line 117 – do you mean “ more severely thermally polluted”?

Line 147 – “(a) our changing climate..” Does this refer to current climate change, rather than predicted ? Perhaps add “current” to clarify

Line 157 “heat recycling is likely feasible “ sounds grammatically wrong, suggest putting “likely feasible” as italics to make clear that this refers to a category (suggest same for mention of categories in other parts of the text)

Line 170 SSPs are mentioned in Figure 5 but not explained in main text. Perhaps a short explanation could be added here.

Line 179 / 180 “increased rate of waste flow” - I would expect that this increase is mainly due to increased natural flow (due to higher air temperatures) while waste heat flows reduce because there is less heat loss from buildings?

Line 223 I would say that “the most sustainable form” is preventing heat losses in the first place, i.e. through better insulation

Line 228 “commonly extracted” – what is meant by this ?

Line 524 + line 583 Needs some mention here of the scale of error / uncertainty associated with thermal property assignment

Line 583 Has saturation status of sediment been considered in thermal properties, i.e. assigning different thermal conductivity values to saturated / unsaturated zone?

Figure 1: add x-axis label

Figure 3: check caption – error in line 6 “for the of our”. Also SSPs not explained in main text - needs some explanation somewhere (as per earlier comment)

Reviewer #3 (Remarks to the Author):

The reviewed manuscript provides a macro-scale resource assessment of thermal disturbances due to urban heat islands. The paper is definitely well written and structured, and based on a sound scientific approach. I believe the manuscript contains some novel and useful elements that merit to be published as a communication in Nature. However, I have two main concerns (see Major Comments below) and believe the manuscript would require further work before publication.

Major Comments

1-Discussion on resource recovery: I am fully aware that the manuscript provides a macro-scale resource assessment and as such the underlying assumption is full resource recovery. My main concern is the fact that the manuscript does not mention nor discuss the fact that no more than 25% of the resource is probably recoverable. Indeed, most unconsolidated sediments are not sufficiently permeable to support the operation of open-loop systems and exploit completely the resource

available. Similarly, closed-loop boreholes can barely exploit it in urbanized environments due to the lack of land area to install them.

The concept of resource is used in the title and abstract but is hardly used clearly thereafter. Perhaps the authors could add a discussion section where they could elaborate more on the concept of resource, but also on the concepts of proven and unproven resource. Authors should also be clear about the limitations of their study in the Abstract and Conclusion to avoid leading policy makers and stakeholders to a technical dead end. Finally, this is a personal and somewhat superficial comment, but I found the concept of "heat recycling" and "waste heat" quite annoying and suggest considering alternatives.

2-Conclusion: A few sentences in the conclusions section are either unnecessarily alarmist, too strong, or simply not supported by the data presented in the study. For example, in P10L218, the statement that most sites are "thermally polluted" is too strong compared to the results shown in Figure S6. Indeed, most of the temperature anomalies presented in Figure S6 are less than 1 or 2 degrees Celsius, which is comparable to local temperature variations. The assertion in P10L215 that "Underground waste heat ... now threatens groundwater quality and groundwater dependent ecosystems" is, in my opinion, too strong and should be lessened. Finally, although I would like to see it, I must admit that I fully disagree with the assertion on P11L262 that the "the feasibility of heat recycling will increase tremendously over time." In regards to my first comment, I suggest to rework substantially the conclusion.

Some additional minor comments and typos

- a) P4L63-64: The analysis of Figure 2A and the associated results are erroneous. Indeed, from Figure 2A I found the 90th percentile equal to 2.3 and 3.2 MJm^{-3} and not 2.6 and 3.5 MJm^{-3} as indicated in the manuscript. I found however that the analysis of the other figures were adequate.
- b) P9L197: Please, add a % symbol after 22 and 7.
- c) P11L239: Please, correct shareholder by stakeholders.
- d) Figure 1: Linking the notion of heat exchange in Figure 1 to eq. 4 should help understanding the general methodology used in the body of work.
- e) References: The references should be checked and corrected to ensure that capital letters are used for City/Country names.

Response to Reviewers

of the manuscript "Shallow subsurface heat recycling: a sustainable global space heating alternative"

by Benz et al

submitted to *Nature Communications*

We have included the reviewers' original comments in black, with our responses interleaved in blue, and denoted by "**Response:**".

Reviewer #1 (Remarks to the Author):

Dear authors,

the manuscript presented by Benz et al. entitled "Shallow underground waste heat: a sustainable global heating resource" for Nature Communications represents a very timely and global assessment that promotes the use of currently wasted heat in conjunction with underground urban heat islands. To my opinion the manuscript only requires minor revisions, here my suggestions:

Response: Thank you Jannis for your favorable and helpful comments! We have addressed them individually in the text below.

- Abstract: specify "other disturbances"; Provide more values for the potential in terms of heat demand today and in the future.

Response: We agree that the term "other disturbances" is not precise enough. Due to the word limit we are not able to provide examples, and instead opted to rephrase the sentence:

"Despite the global interest in green energy alternatives, little attention has focused on the large-scale viability of recycling the ground heat accumulated due to urbanization, industrialization and climate change." [Abstract]

- p. 2, l. 25: specify: "potential consequences"

Response: We have restructured the paragraph to read:

"and (4) not recycling the accumulating heat will result in further underground thermal pollution with potential consequences such as declining groundwater quality (22-24) and adverse effects on groundwater-dependent ecosystems (25,26)." [Line 25]

[22] Danielopol, D. L., Griebler, C., Gunatilaka, A. & Notenboom, J. Present state and future prospects for groundwater ecosystems. *Environmental Conservation* 30, 104–130 (2003). URL <https://doi.org/10.1017/s0376892903000109>.

[23] Bonte, M., Stuyfzand, P. J., van den Berg, G. A. & Hijnen, W. A. M. Effects of aquifer thermal energy storage on groundwater quality and the consequences for drinking water production: a case

study from the netherlands. *Water Science and Technology* 63, 1922–1931 (2011). URL <https://doi.org/10.2166/wst.2011.189>.

[24] Riedel, T. Temperature-associated changes in groundwater quality. *Journal of Hydrology* 572, 206–212 (2019). URL <https://doi.org/10.1016/j.jhydrol.2019.02.059>.

[25] Kurylyk, B. L., MacQuarrie, K. T. B., Linnansaari, T., Cunjak, R. A. & Curry, R. A. Preserving, augmenting, and creating cold-water thermal refugia in rivers: concepts derived from research on the Miramichi river, New Brunswick (Canada). *Ecohydrology* 8, 1095–1108 (2014). URL <https://doi.org/10.1002/eco.1566>.

[26] Koch, F. et al. Groundwater fauna in an urban area – natural or affected? *Hydrology and Earth System Sciences* 25, 3053–3070 (2021). URL <https://doi.org/10.5194/hess-25-3053-2021>.

- p. 3, l. 49: already define "natural levels" here

Response: After noticing that we only use the term natural levels at three places in the manuscript, we replaced it with the unambiguous term “undisturbed levels” at all locations. [Lines 51, 53, 153 and caption Fig 1]

- p. 10, l. 212-214: include in abstract?

Response: We agree that this is one of the main benefits of heat recycling and worth focusing on. However, unfortunately we are at the absolute maximum word count in the abstract already and have hence decided to not include this sentiment in the abstract.

- Generally: Discuss and differentiate between theoretical and technical (and further?) potentials.

Response: Thank you for the suggestion. We agree and have added some sentences highlighting the importance of the technical potential (i.e. the accessibility of the ground for geothermal systems) when extracting this energy. We believe in-depth discussion of further potentials (considering for e.g. economics) is beyond the scope of this study.

In the introduction we specify:

“The objective of this study is to quantify the theoretical feasibility of subsurface heat recycling at a multi-continental scale, focusing on its energetic sustainability and renewability under present and future climate conditions without consideration of any technical constraints.” [Line 43]

In the results section potential technical constraints are again highlighted:

“We note that even in these unconsolidated sediments local conditions may vary, and not all heat is recoverable based on today’s technology and land availability.” [Line 84]

Lastly, we highlight them again in the summarizing discussion:

“Like the thermal pollution in subsurface urban heat islands, warming climates at the surface will contribute to underground heat reservoirs where the excess heat can be recycled with shallow geothermal energy systems. The ability to utilize this theoretical geothermal potential will depend on the accessibility and extractability of the ground heat for geothermal systems.” [Line 232]

- Figure 2/4/5: Enlarge world map and include zooms; enlarge a,b,c...

Response: Due to the limited space available we decided to instead add the maps shown in figures 2, 4, and 5a to the google earth engine app we have so far used to display the results of Figure 5b.

We refer to the site at the appropriate places in the figure captions:

“Maps displaying the accumulated heat. A zoomable version of this map is accessible under <https://susanneabenz.users.earthengine.app/view/feasible-heat-recycling>.” [caption Fig. 2]

“Maps of the annual mean heat exchange for the *recycled* scenarios. A zoomable version of this map, as well as the map for the scenarios under *climate change* are accessible under <https://susanneabenz.users.earthengine.app/view/feasible-heat-recycling>.” [caption Fig. 4]

“Results of the random forest classification. A zoomable version of these maps is available from <https://susanneabenz.users.earthengine.app/view/feasible-heat-recycling>.” [caption Fig. 5]

- Figure 5: It seems that there is a strap in northern Netherlands / Germany / Poland, is this an artifact?

Response: Yes, it is an artifact. While we initially just accepted the anomaly as an artifact of the random forest, we have now spent some additional time testing out other options. Based on these tests we have learned that the classification used in the manuscript so far was over-reliant on latitude. By including latitude in the training data, the algorithm “learned” that locations at a similar latitude as Northern German, where groundwater levels are shallow and heat input is therefore high, also have a high heat input.

Hence we have decided to run the random forest without Latitude, which was initially included as a proxy for undisturbed groundwater temperatures.

This decision is based on two points:

a) While the artifact shown on the map had no real implications for our finding as it misclassified regions with a very low population density, it still resulted in a misleading image that implied more use of heat recycling than appropriate.

b) Latitude is not the best proxy for undisturbed groundwater temperatures, particularly in regions with an Alpine environment (such as central Europe). While we considered using air temperature as a proxy instead, we have decided against this as they are also disturbed in an urban environment.

Outside of the artifact you described, the new classification and the old classification give very similar results. The accuracy is just slightly reduced from

0.83 to 0.82. Training data is now also selected based on groundwater table and not based on latitude.

The following changes in text were necessary to implement this new classification:

In the results section the description of our findings now reads:

“The overall accuracy based on the confusion matrix of validation data is 0.81 (Fig. S5). In total 350,000 km² were classified. Of those, 90% (housing 74% of the population) are classified as *not feasible*, 2% (housing 23% of the population) as *potentially feasible*, and 7% (housing 2% of the population) as likely feasible.” [Line 217]

The summarizing discussion was reworded to the following:

“Based on our estimate, 7% of the European study area can supply more than 25% of the annual heating demand by recycling the annual heat input. An additional 2% of the area (housing 23% of the European population) could generate up to 25% of the annual heating demand.” [Line 251]

The method section changed in three places:

“Summarizing variables as much as possible we use heating demand (determining it per pixel following eq. 9), building density (following eq. 3), depth to the groundwater table, thermal conductivity, and ground surface temperature to train the classifier. Histograms of all variables are displayed in Figure S9.” [Line 505]

“Considering the importance of training data sample selection, we assign probabilities based on the depth to the groundwater table at our locations - the variable with a distribution least represented by the data set. Probabilities are thus set to the ratio of the histograms for groundwater table depth of pixels and groundwater table depth of point locations. Figure S9 displays the cumulative histograms for all variables for all locations as well as for the selected training locations.” [Line 513]

“Overall accuracy based on the training data is 0.96, while overall accuracy based on the validation data is 0.81.” [Line 521]

Additionally Figures 5, S5 and S9 have been updated with the new results.

Fig 5: **Feasibility of subsurface heat recycling for our recycled scenario.** Please note: for all parts of this figure only locations with unconsolidated sediments are shown, or rather classified. **a:** Map displaying the feasibility of recycling the annual heat input. **b:** Results of the random forest classification. A zoomable version of these maps is available from <https://susanneabenz.users.earthengine.app/view/feasible-heat-recycling>.

Fig S5: **Accuracy assessment of the random forest classification.** The y-axis shows the observed ratio of generated thermal energy when recycling the annual heat input and heating demands, while the x-axis indicates classification results. Green bars represent all locations used for training, blue bars locations used for validation only.

Fig S5: **Distribution of variables in training data and pixels to be classified.** Cumulative histograms of all variables used for the random forest classification for all European locations with unconsolidated sediments (orange line), the excerpt of those used to train the random forest classifier (red line), and all pixels on which the classifier has been applied to (black line).

All the best,
Jannis Epting

Reviewer #2 (Remarks to the Author):

This paper presents a global approach for estimating shallow subsurface heat resources, now and in the future, and assesses where they have utility to cover current and future heat demand. While global assessment come with a great set of uncertainties and caveats (one of which being that they cannot be validated), they have value in highlighting the scale of available resource potential.

The methodology draws on a number of existing approaches developed by the authors, which have been combined and extended in this paper for application on the global scale. While global assessments have been done for related resources, e.g. for deep geothermal heating/ cooling, shallow heat resources have not been quantified on the global scale. As such the approach is novel and will be of interest to the readers of this journal.

The paper is very well written and the methodology is explained in sufficient detail to make it reproducible. I have some comments for consideration by the authors, as detailed below:

Response: Thank you very much for the kind review and the helpful comments! We have addressed all of them individually in the following.

(1) The paper is very well written, but having methodology at the end means that, when reading main body of this paper, reader is left wondering in some places how parameters were derived/ what was/wasn't considered etc. This can get in the way of understanding what figures show and what the finding actually mean in practical terms. I suggest to add some very brief explanations to the text, and have included some suggestions below where I think this would improve readers' understanding.

Response: We understand your concern. We are now giving more detail on the input parameters and processes considered in our methods at the appropriate places of the results section.

We already included this information for the accumulated heat:

“In many locations worldwide, shallow subsurface temperatures are elevated due to accumulated heat from infrastructure, climate change, and land cover/ land use changes. We quantified the volumetric density of excess thermal energy (Fig. 2) as the product of heat anomalies (the difference between local groundwater temperatures and median rural background groundwater temperatures) and ground volumetric heat capacity (see Methods for a detailed description of parameters and equations).” [Line 59]

We now also give more information on estimating heating demand, the heat generated from heat recycling, and annual heat exchange, and the random forest classifier:

“To quantify the feasibility of subsurface heat recycling, we compare the local annual space heating demands, estimated from heating degree days, population density, and GDP (34), to the accumulated heat (expressed as a volumetric density) in Figure 3A.” [Line 88]

“Diagonal lines indicate the necessary vertical extraction interval to meet one year's heating demand through the energy generated when recycling the accumulated

heat with a geothermal system and heat pump with a COP of 3.5” [Line 91]

“The annual heat exchange calculated from groundwater, surface and building temperatures and considering groundwater table depth and building density, shows that for 79% (16%, 97%) of the more than 8,000, mainly rural locations ... ” [Line 116]

“We will now expand on the results of our sites to map the feasibility of heat recycling in Europe where more than 90% of our sites are located. For this analysis we expand on the scenario *recycled* which identifies suitable locations for present-day sustainable and renewable subsurface heat recycling (Fig. 5A). This is achieved by training a random forest classifier on 70% of all suitable locations based on heating demand, building density, depth to the groundwater table, thermal conductivity, and ground surface temperatures.” [Line 209]

[34] Isaac, M. & van Vuuren, D. P. Modeling global residential sector energy demand for heating and air conditioning in the context of climate change. *Energy Policy* 37, 507–521 (2009). URL <https://www.sciencedirect.com/science/article/pii/S0301421508005168>.

(2) There is some confusion as to when the analysis was done on the full data sets (>8000) and when it used the subset (unconsolidated sediments) which should be clarified. See comment in detailed list below. If different data sets were used for the different analyses, it would be useful if this was highlighted in the figure captions too.

Response: Thank you for the comment! Following your advice below we adjusted our text to mark more clearly when the full dataset and when the subset with unconsolidated sediments is used.

We have added a clarifying sentence when we first introduce the subset:

“In this study, we use the subset of locations with unconsolidated sediment whenever we compare our results to heating demands, but work with the entire data set when quantifying accumulating heat.” [Line 86]

For further clarification we have added the following when comparing heating demands to annual heat input:

“Following our analysis of accumulated waste heat, we quantify the feasibility of recycling this heat by comparing heating demands and the annual heat input (Fig. S4) for the subset of data with unconsolidated sediments.” [Line 133]

“By comparing local heat input (i.e., heat input once the accumulated heat has been recycled) to heating demands (Fig. 3B) for our subset of data in unconsolidated sediments, we find that...” [Line 170]

“Heat input is then compared to space heating demands for the same time period (2095-2099) for our subset of data in unconsolidated sediments.” [Line 193]

We have also marked this more clearly in the caption of Figures 3 and 5, and supplementary figures S2 and S4 which are the only figures relying on the subset with unconsolidated sediments.

“Figure 3: Feasibility of subsurface heat recycling under all scenarios. Please note: for all parts of this figure only locations with unconsolidated sediments are shown. **a:** Comparison of accumulated waste heat...” [Caption Fig 3]

“Figure 5: Feasibility of subsurface heat recycling for our recycled scenario. Please note: for all parts of this figure only locations with unconsolidated sediments are shown, or rather classified. **a:** Map displaying the...” [Caption Fig 5]

“Figure S2: Extraction interval and borehole depth needed to supply one year of heat from accumulated heat. This analysis is based only on locations with unconsolidated sediments.” [Caption Fig S2]

“Figure S4: Comparison of annual mean heat input of our scenario *status quo* and heating demand. Only locations with unconsolidated sediments are shown.” [Caption Fig S4]

(3) Given that this is a global assessment, the uncertainties associated with assessment itself as well as linked to assumptions/ inherent in the data sets could have been discussed in a bit more detail and how they impact the resource estimates. Uncertainties have been mentioned for some of the data sets/parameters, but not for others. I have included examples in list below of where I think this could be added, e.g. for thermal property data.

Response: We agree. Sofar uncertainties were primarily given in the figures, we now also give numbers for our minimum and maximum estimates for all appropriate locations in the text:

This was already noted partially when quantifying accumulated heat:

“and we find accumulated heat up to 2.8 MJm⁻³ (90th percentile, 2.5 MJm⁻³) based on our lowest estimates of input parameters and 3.2 MJm⁻³ based on the highest estimates; Fig. 2A). For the remainder of this study all results will be given in the form *results main analysis (results lowest estimates, results highest estimates)*. After weighting for population density, we find that 71% (68%, 75%) of people live where the study sites that have accumulated shallow subsurface heat, and 26% (27% following highest and lowest estimates of heat capacity and population densities) live where accumulated heat is above the 90th percentile for the total dataset (Fig. 2A).” [Line 65]

We have now added uncertainties to for relevant numbers where not already given:

“For example, 12% (9%,15%) of the sites are located below the 1 m line, which means that recycling the accumulated heat of an only 1 m thick zone of permeable ground could supply at least an entire year's worth of heat” [Line 93]

“For 50% (47%, 52%) of all locations an extraction interval of 20 m or less will suffice to supply one year or more of heat (for 4% (39%, 47%) 10 m are sufficient; Fig. S2A). However, as the extraction interval should be located in the aquifer, the required borehole length needs to be longer (extraction interval plus depth to groundwater table, Fig. S2B). Still, for 43% (0%, 49%) of all locations a borehole

length of 20 m can generate enough heat for one year; for 29% (0%, 41%) 10 m will suffice. The high uncertainties are a product of the high uncertainties in the depth to the groundwater table, where we assume values of ± 10 m for our low and high estimates.” [Line 98]

“The annual heat exchange calculated from groundwater, surface and buildings temperatures and considering groundwater table depth and building density, show that for 79% (97%, 16%) of the more than 8,000, mainly rural locations, the subsurface presently loses heat to the surface; however, values overall range from -56.5 to $+2.7$ MJm^{-2} (10th - 90th percentile; -158.8 to -5.5 MJm^{-2}) for the lowest estimates of input parameters and -0.8 to 32.7 MJm^{-2} for the highest estimates; Fig. 4A). The high uncertainties here and in the following stem again from uncertainties associated with the depth to the groundwater table.” [Line 116]

“We sort locations into *not feasible* (no heat input, i.e., heat loss, or no heating demand -- 82% of all locations in unconsolidated soils), *potentially feasible* (generated heat from recycling the heat input is less than or equal to one quarter of the annual heating demand -- 4%), or *likely feasible* (generated heat from recycling the heat input is larger than one quarter of the annual heating demand -- 14%). Based on our lowest estimates of input parameters 99% of locations are *not feasible* due to the assumed distance to the groundwater table. Based on our highest estimates 14% are *not feasible*, 7% are *potentially feasible*, and 78% are *likely feasible*.” [Line 134]

“In this scenario, aquifers in 28% (3%, 91%) of the locations (compared to 21% (3%, 84%) in the *status quo*) have heat input from buildings and/or elevated surface temperatures [...] and heat exchange is overall higher, ranging from -37.0 to $+4.1$ MJm^{-2} (10th - 90th percentile; -125.8 to -4.5 MJm^{-2}) for the lowest estimates of input parameters and $+0.2$ to 35.4 MJm^{-2} for the highest estimates; Fig. 4A).” [Line 154]

“By comparing local heat input (i.e., heat input once the accumulated heat has been recycled) to heating demands (Fig. 3B) for our subset of data in unconsolidated sediments, we find that for 18% of all locations (housing 10% of the population), subsurface heat recycling is *likely feasible*, i.e., the heat generated when extracting the annual heat input is more than one quarter of annual space heating demands. For another 7% (housing 20% of the population), subsurface heat recycling is *potentially feasible* (i.e., the heat generated is between 0 and 25% of the heating demand), whereas the remaining locations have either no heat input or no heating demand. In our minimum estimates, 99% of locations are *not feasible* and 1% *potentially feasible*; in our maximum estimates 6% are *potentially feasible* and 87% *likely feasible* - at least based on current surface temperatures” [Line 170]

“These analyses project that by the end of the century (2095-2099) following RCP 4.5, 85% (4%, 100%) of all our locations (99% (72%, 100%) in RCP 8.5) will have heat input to the subsurface from increased surface temperatures and/or buildings, and overall ranges are elevated to -5.2 to $+49.0$ MJm^{-2} (-155.0 to -3.7 MJm^{-2}), $+7.0$ to $+64.0$ MJm^{-2}) for RCP 4.5 and $+6.7$ to $+125.7$ MJm^{-2} (-49.2 to $+4.3$

MJm⁻², +20.5 to +476.9 MJm⁻²) for RCP 8.5 (10th to 90th percentile, Fig. 4B).” [Line 187]

“Following RCP 4.5, only 16% (98%, 0%) of our locations are *not feasible* for subsurface heat recycling, 2% (1%, 0%) are *potentially feasible*, and 81%(1%, 100%) are *likely feasible* (i.e., the heat generated when sustainably extracting the heat input is more than one quarter of space heating demands). In fact, for 73% (0%, 98%) of our locations the generated recycled heat is equal to or larger than local heating demands. For RCP 8.5, 99% (53%, 100%) of all locations are *likely feasible* with 97% (30%, 100%) projected to generate at least as much heat as the annual space heating demands when extracting the annual heat input.” [Line 198]

We have also followed your suggestions from the specific comments below. That is, we now describe in detail how uncertainties for the thermal property data have been chosen. Fortunately thermal property variation in geologic materials is relatively constrained compared to for example hydraulic properties.

“... Here, we assign to each class heat capacities based on the VDI 4640 guidelines (Table S1, Fig. S7). These guidelines give a range of values for each soil type. We use the minimum and maximum of that range for our minimum and maximum analysis, and the mean of these two values for our main analysis. Where our soil types consist of two rock types described in the VDI 4640, we take the mean of these rock types. Furthermore, for unconsolidated sediments we use the values given for the water-saturated respective soil type to best represent conditions in the aquifer.” [Line 325]

“The process to determine thermal conductivity is analogous to the one determining heat capacity (Table S1, Fig. S7). However, here the VDI 4640 guidelines give recommended values which we use in our main analysis. For our minimum and maximum analysis we still rely on the ranges given in the guidelines. As heat transport is through the unsaturated zone, for unconsolidated sediment we use values recommended for a moist environment, or where these are not given, the mean of the recommendations for dry and water-saturated conditions.” [Line 383]

Specific comments:

Line 79 “unconsolidated sediments” could be understood to include clays etc. I suggest to be more specific here, i.e. ...rock environments, we only include locations where unconsolidated sediments of suitable grainsize are present (leaving”

Response: We agree and have added some clarification:

“filter out all locations without unconsolidated sediments (i.e. keeping only locations with the soil types *sand and coarser, sand and silt or clay, silt and/or clay, or mixed* following Table S1; leaving us with just...” [line 82]

Line 101 need to make clear here that this sentence refers to references in previous sentence , perhaps add “...garages were found by thee studies to”

Response: We have adjusted the paragraph for clarification. It now reads:

“These fluxes have previously been determined for selected cities (14, 19, 35, 36). In three of these studies local infrastructure such as district heating networks and subsurface parking garages dominate on the smaller scale. However, over larger areas the studies indicate that buildings and elevated temperatures caused by sealed surfaces are dominant contributors to subsurface heat accumulation” [Line 109]

[14] Benz, S. A., Bayer, P., Menberg, K., Jung, S. & Blum, P. Spatial resolution of anthropogenic heat fluxes into urban aquifers. *Science of The Total Environment* 524-525, 427–439 (2015). URL <https://doi.org/10.1016/j.scitotenv.2015.04.003>

[19] Tissen, C. et al. Identifying key locations for shallow geothermal use in Vienna. *Renewable Energy* 167, 1–19 (2021). URL <https://doi.org/10.1016/j.renene.2020.11.024>

Menberg, K., Blum, P., Schaffitel, A. & Bayer, P. Long-term evolution of anthropogenic heat fluxes into a subsurface urban heat island. *Environmental Science & Technology* 47, 9747–9755 (2013). URL <https://doi.org/10.1021/es401546u>.

[36] Benz, S. A. et al. Comparing anthropogenic heat input and heat accumulation in the subsurface of Osaka, Japan. *Science of The Total Environment* 643, 1127–1136 (2018). URL <https://doi.org/10.1016/j.scitotenv.2018.06.253>.

Line 105 this is where I think a bit more detail would be useful , e.g. adding something like “The annual heat exchange calculated from groundwater, surface and buildings temperatures and considering groundwater table depth and building density, show that for 79%”

Response: We agree with the suggested changes and have implemented them.

“The annual heat exchange calculated from groundwater, surface and buildings temperatures and considering groundwater table depth and building density, show that for 79% (16%, 97%) of the more than 8,000, mainly rural locations ... ” [Line 116]

Line 105 “more than 8,000 ,,locations” – was this analysis done on the full data set rather than the unconsolidated sediment subset?

Response: Yes, this analysis was done for all locations. As responded to your comment above we only focused on the unconsolidated sediment subset when comparing to heating demand which is now stated more clearly in the manuscript.

Line 114 -116 “we find no reverse correlation herehave reached their limit” – Could this be due to lower air temperatures in rural areas? I.e. even though waste heat accumulation is much lower in rural areas so are air temperatures, i.e. T gradients and hence fluxes could be quite similar to urban areas?

Response: Thank you for pointing this out. This is definitely also a reason for the high amount of heat thermally polluted aquifers transport to the surface in rural regions. We have adjusted the sentence to include this sentiment:

“...implying that it is disproportionately the more suburban or even rural thermally polluted groundwater reservoirs that have reached their limits and generate heat flow to the (comparatively) cold surface. While these are less common than heated urban aquifers, ...” [Line 128]

Line 117 – do you mean “ more severely thermally polluted”?

Response: You are correct. we added the term “thermally” into the sentence.

“While these are less common than heated urban aquifers, they have been identified as more thermally polluted in previous studies” [Line 130]

Line 147 – “(a) our changing climate..” Does this refer to current climate change, rather than predicted ? Perhaps add “current” to clarify

Response: Again correct, we were referring to changing temperatures in the past (or rather still ongoing). We have adjusted the sentence to make this more clear:

“(a) the rising temperatures of the past decades lead to heat input in places outside the built environment” [Line 161]

Line 157 “heat recycling is likely feasible “ sounds grammatically wrong, suggest putting “likely feasible” as italics to make clear that this refers to a category (suggest same for mention of categories in other parts of the text)

Response: We see your concern and followed the suggestion to mark the names of the categories in italics in the main manuscript, in figure captions, in the method section and in the supplement.

Line 170 SSPs are mentioned in Figure 5 but not explained in main text. Perhaps a short explanation could be added here.

Response: We’ve now included the shared socioeconomic pathways in the main part of the manuscript:

“In addition to the impact of increased temperatures, these demands are also adjusted for projected population densities following the shared socioeconomic pathways (SSPs) (37,38).” [Line 194]

[38] Jones, B. & O’Neill, B. Global population projection grids based on shared socioeconomic pathways (SSPs), downscaled 1-km grids, 2010-2100 (2019). URL <https://sedac.ciesin.columbia.edu/data/set/popdynamics-pop-projection-ssp-downscaled-1km-2010-2100>.

[39] Gao, J. Downscaling global spatial population projections from 1/8-degree to 1-km grid cells (2017). URL <https://opensky.ucar.edu/islandora/object/technotes:553>

Line179 / 180 “increased rate of waste flow” - I would expect that this increase is mainly due to increased natural flow (due to higher air temperatures) while waste heat flows reduce because there is less heat loss from buildings?

Response: You are correct. Reviewer 3 also pointed out that waste heat is a misleading term when talking about heat input from climate change. We have thus adjusted our wording here and throughout the manuscript to ensure that the term “Waste heat” is only used when appropriate - and not e.g. in this instance.

“We project that the feasibility of subsurface heat recycling will increase drastically

(Fig. 3C and D) due to the combined effects of increased rate of heat flow from the surface and decreased heating demands in a warmer world.” [Line 196]

Line 223 I would say that “the most sustainable form” is preventing heat losses in the first place, i.e through better insulation

Response: We agree that not accumulating heat from buildings or climate change (rendering heat recycling unnecessary) would be the best option. We address this briefly in the manuscript a bit earlier than you suggested..

“If we do not continuously reduce or recycle the annual heat input, locations with a positive heat exchange in this *recycled* scenario would begin accumulating heat again.” [Line 166]

We have now also added a reference to preventing heat accumulation in the summarizing discussion section:

“If we cannot block (e.g., through insulation) the annual subsurface heat input from the surface and buildings, heat will continue to accumulate, unless it is continuously recycled. Based on our estimate...” [Line 248]

Line 228 “commonly extracted” – what is meant by this ?

Response: We have rephrased the sentence for clarification:

“It is important to note that this is in addition to the extractable geothermal heat flux that is...” [Line 2251]

Line 524 + line 583 Needs some mention here of the scale of error / uncertainty associated with thermal property assignment

Response: We have added a more detailed explanation of the uncertainties assumed in our analysis:

“... Here, we assign each class heat capacities based on the VDI 4640 guidelines (Table S1, Fig. S7). These guidelines give a range of values for each soil type. We use the minimum and maximum of that range for our minimum and maximum analysis, and the mean of these two values for our main analysis. Where our soil types consist of two rock types described in the VDI 4640, we take the mean of these rock types. Furthermore, for unconsolidated sediments we use the values given for the water-saturated soil type to best represent conditions in the aquifer.” [Line 330]

“The process to determine thermal conductivity is analogous to the one determining heat capacity (Table S1, Fig. S7). However, here the VDI 4640 guidelines give recommended values which we use in our main analysis. For our minimum and maximum analysis we still rely on the ranges given in the guidelines.” [Line 387]

Line 583 Has saturation status of sediment been considered in thermal properties, i.e assigning different thermal conductivity values to saturated / unsaturated zone?

Response: Yes, the saturation status has been considered. As we calculate the heat transport from the surface to the groundwater table, i.e. through the unsaturated zone, we only consider thermal conductivity of the unsaturated zone. The precise details have now been added:

“As heat transport is through the unsaturated zone, for unconsolidated sediment we use values recommended for a moist environment, or where these are not given, the mean of the recommendations for dry and water-saturated conditions.” [Line 386]

Figure 1: add x-axis label

Response: The x-axis represents location, we have added some clarifying text.

Figure 3: check caption – error in line 6 “for the of our”. Also SSPs not explained in main text - needs some explanation somewhere (as per earlier comment)

Response: Thank you for noticing! the error was removed:

“Comparison of annual mean heat input and heating demand for the scenario *recycled*.” [Caption Figure 3]

As per earlier comment, a short introduction to SSPs is now included in the main text.

Reviewer #3 (Remarks to the Author):

The reviewed manuscript provides a macro-scale resource assessment of thermal disturbances due to urban heat islands. The paper is definitely well written and structured, and based on a sound scientific approach. I believe the manuscript contains some novel and useful elements that merit to be published as a communication in Nature. However, I have two main concerns (see Major Comments below) and believe the manuscript would require further work before publication.

Response: Thank you very much for your constructive feedback! We have addressed all of your concerns individually below.

Major Comments

1-Discussion on resource recovery: I am fully aware that the manuscript provides a macro-scale resource assessment and as such the underlying assumption is full resource recovery. My main concern is the fact that the manuscript does not mention nor discuss the fact that no more than 25% of the resource is probably recoverable. Indeed, most unconsolidated sediments are not sufficiently permeable to support the operation of open-loop systems and exploit completely the resource available. Similarly, closed-loop boreholes can barely exploit it in urbanized environments due to the lack of land area to install them.

Response: We understand and agree with these concerns. The present study focuses on quantifying the theoretical potential and not the technical potential (i.e. operational capacity for ground heat recycling). The reviewer is correct that this overestimates the technical potential, which may be in a range of up to 50% of the theoretical potential based on previous work on shallow geothermal energy application. Still, with our work we address the potential of “full” waste heat recovery. In order to reach this, there are several ongoing research initiatives to support spatial geothermal system planning in highly used urban areas. The improvements that can be expected from these works will contribute to continuously enhancing the recovery of the geothermal resource in the future. In addition, the reviewer correctly points out that permeability is important and accordingly our analysis only discusses unconsolidated sediments when comparing the accumulated heat or heat input to heating demands (i.e. implying it is recoverable). We disagree with the statement that most unconsolidated sediments are not suited for open-loop systems. They rely on groundwater availability and at least a moderate aquifer permeability, and this is found in most regions with shallow unconsolidated formations.

We now note the (current) technical constraints of heat recycling more clearly in the manuscript at three locations (Introduction, Results, and Discussion) to address the concerns of reviewers 1 and 3:

“The objective of this study is to quantify the theoretical feasibility of subsurface heat recycling at a multi-continental scale, focusing on its energetic sustainability and renewability under present and future climate conditions without consideration of any technical constraints.” [Line 43]

“We note that even in these unconsolidated sediments local conditions may vary, and not all heat is recoverable based on today's technology and land availability.” [Line 84]

“Like the thermal pollution in subsurface urban heat islands, warming climates at the surface will contribute to underground heat reservoirs where the excess heat can be recycled with shallow geothermal energy systems. The ability to utilize this theoretical geothermal potential will depend on the accessibility and extractability of the ground heat for geothermal systems. Subsurface heat recycling will allow us to capitalize on warming climates, while actively mitigating future climate change through a low-carbon heating approach and reducing the adverse effects of rising

temperatures on our (ground-)waters” [Line 232]

The concept of resource is used in the title and abstract but is hardly used clearly thereafter. Perhaps the authors could add a discussion section where they could elaborate more on the concept of resource, but also on the concepts of proven and unproven resource. Authors should also be clear about the limitations of their study in the Abstract and Conclusion to avoid leading policy makers and stakeholders to a technical dead end. Finally, this is a personal and somewhat superficial comment, but I found the concept of "heat recycling" and "waste heat" quite annoying and suggest considering alternatives.

Response: We agree that we must avoid leading policy makers and stakeholders to a technical dead end. Hence we have followed your advice and have refrained from using the term “resource” in favor of “theoretical potential”. We do not believe that the concepts of proven and unproven potential can be used at the here discussed scale - whether this energy can be recovered (and hence is a proven or unproven resource) is highly dependent on location. Other changes noted in response to the previous comment will also help policy makers understand we are quantifying theoretical potential.

We have changed the title to: “Shallow subsurface heat recycling: a sustainable global space heating alternative”

We have adjusted the abstract in two locations to remove the term resource and better manage expectations:

“We investigate this theoretical heat potential at a multi-continental scale by first leveraging datasets of groundwater temperature and lithology...”

“Results highlight that subsurface heat recycling warrants consideration in the move to a low-carbon economy in a warmer world.”

Similarly we have removed the term “resource” in the result section in one location.

“Hence, extracting the annual heat input each year would not only be the most sustainable and renewable use of this heat, it is also the only approach” [Line 168]

We understand that the term heat recycling is used by many and as such means different things in different scientific disciplines. We feel it is warranted here as we focus on additional heat available that stems from anthropogenic sources and/or climate change. Since it would be lost/ignored otherwise, our proposal is to recycle it rather than to deposit it. A definition is now also included in the abstract and the introduction. Additionally, in order to differentiate our work from other uses of the term we have changed the term to “subsurface heat recycling” where appropriate throughout the manuscript.

The abstract now states:

“Despite the global interest in green energy alternatives, little attention has focused on the large-scale viability of recycling the ground heat accumulated due to urbanization, industrialization and climate change.”

A definition was also added to the introduction:

“The relative lack of attention paid to large-scale subsurface heat recycling (i.e., extraction of the additional heat available in the shallow subsurface that stems from urbanization, industrialization and climate change) as a green energy solution with potential for global climate change mitigation is presumably due...” [Line 17]

Lastly, we believe the term “waste heat” describes quite adequately the heat input from anthropogenic infrastructure into the underground, but we agree that it might not be the correct term to use for heat input from climate change. We hence double checked each use of the term and adjusted it throughout the manuscript when not explicitly referring to heat input from buildings or other underground infrastructure.

2-Conclusion: A few sentences in the conclusions section are either unnecessarily alarmist, too strong, or simply not supported by the data presented in the study. For example, in P10L218, the statement that most sites are “thermally polluted” is too strong compared to the results shown in Figure S6. Indeed, most of the temperature anomalies presented in Figure S6 are less than 1 or 2 degrees Celsius, which is comparable to local temperature variations. The assertion in P10L215 that “Underground waste heat ... now threatens groundwater quality and groundwater dependent ecosystems” is, in my opinion, too strong and should be lessened. Finally, although I would like to see it, I must admit that I fully disagree with the assertion on P11L262 that the “the feasibility of heat recycling will increase tremendously over time.” In regards to my first comment, I suggest to rework substantially the conclusion.

Response: We understand your concern and in part agree.

We disagree that the term “thermally polluted” is too strong. It is used in the subsurface heat island and geothermal literature quite frequently even for seemingly small temperature anomalies (e.g. Blum et al. 2021). We also note that the seasonal temperature variation declines with depth, hence even a shift in groundwater temperature of as little as 1°C is much higher than what we would expect naturally. While seemingly small, a permanent shift of this magnitude may interfere with thermosensitive groundwater dependent ecosystems and thus fit the EU definition of pollutants. Not yet a worrisome pollution, but a pollution nonetheless.

We agree that the sentence in line 215 you highlight is too strong and have adjusted it accordingly:

“Over the past few decades heat has accumulated in the underground where it might impact groundwater quality and groundwater dependent ecosystems, including the thermal regimes of surface water bodies with thermal regimes and thermal heterogeneity influenced by groundwater inputs.” [Line 239]

We are unsure what you mean when you say you disagree with the shown increase in feasibility of heat recycling over the decades as this is one of the main findings of our analysis. We assume, similar to your comment above, you are mainly concerned about the term “heat recycling” and how it is used here to describe the recovery of heat caused by climate change? As discussed above we have added a definition on our use of that term in the introduction of the paper and hope you are

now able to agree with our assertion that the feasibility to “recycle global warming” will increase over the coming century. However, we have removed the term “tremendously” in the quote you point out:

“We also showcase the option to recycle the heat input from climate change – in contrast to other common methods of renewable energy supply, the feasibility of subsurface heat recycling will increase over time.” [Line 260]

Philipp Blum, Kathrin Menberg, Fabien Koch, Susanne A. Benz, Carolin Tissen, Hannes Hemmerle, Peter Bayer, Is thermal use of groundwater a pollution? Journal of Contaminant Hydrology, Volume 239, 2021, 103791, <https://doi.org/10.1016/j.jconhyd.2021.103791>.

Some additional minor comments and typos

a) P4L63-64: The analysis of Figure 2A and the associated results are erroneous. Indeed, from Figure 2A I found the 90th percentile equal to 2.3 and 3.2 MJm⁻³ and not 2.6 and 3.5 MJm⁻³ as indicated in the manuscript. I found however that the analysis of the other figures were adequate.

Response: Thank you very much for taking the time and double checking this with data from the repository! We really appreciate it. And you are correct. The numbers in the manuscript still represent a previous version with slightly different values for c_v . They should be 2.5 and 3.2 MJm⁻³ as these are the numbers I get running the code in Python or Excel with the data from the repository or the data still on my drive.

We corrected the values in the manuscript:

“We find accumulated waste heat from 0 to 2.8~\MJqm (90th percentile, 2.5 MJqm based on our lowest estimates of input parameters and 3.2 \MJqm based on the highest estimates” [Line 67]

b) P9L197: Please, add a % symbol after 22 and 7.

Response: We have added the % symbol.

c) P11L239: Please, correct shareholder by stakeholders.

Response: We corrected the term “shareholder” to “stakeholder”.

“Should policymakers and stakeholders decide against shallow geothermal heat recycling this heat will continue to accumulate in the underground adversely affecting water quality and ecosystems.” [Line 268]

d) Figure 1: Linking the notion of heat exchange in Figure 1 to eq. 4 should help understanding the general methodology used in the body of work.

Response: We have linked the caption to Figure 1 to the corresponding equations.

“Figure 1: **Schematic drawing of the analyzed scenarios.** **a:** The *status quo* describes elevated groundwater temperatures (GWTs) under the built environment that result in accumulated heat (see eq. 2). **b:** The scenario *recycled* highlights heat exchange (see eq. 4) between the surface and aquifer...” [Caption Figure 1]

e) References: The references should be checked and corrected to ensure that capital letters are used for City/Country name

Response: We have adjusted our Latex file to capitalize city and country names.